# Nr2f1a maintains atrial *nkx2.5* expression to repress pacemaker identity within venous atrial cardiomyocytes of zebrafish

Kendall E Martin[1,2], Padmapriyadarshini Ravisankar[2], Manu Beerens[3], Calum A MacRae[3], Joshua S Waxman[2,4]*

[1]Molecular Genetics, Biochemistry, and Microbiology Graduate Program, University of Cincinnati College of Medicine, Cincinnati, United States; [2]Molecular Cardiovascular Biology Division and Heart Institute, Cincinnati Children's Hospital Medical Center, Cincinnati, United States; [3]Divisions of Cardiovascular Medicine, Genetics and Network Medicine, Department of Medicine, Brigham and Women's Hospital and Harvard Medical School, Boston, United States; [4]Department of Pediatrics, University of Cincinnati College of Medicine, Cincinnati, United States

*For correspondence:
joshua.waxman@cchmc.org

Competing interest: The authors declare that no competing interests exist.

**Abstract** Maintenance of cardiomyocyte identity is vital for normal heart development and function. However, our understanding of cardiomyocyte plasticity remains incomplete. Here, we show that sustained expression of the zebrafish transcription factor Nr2f1a prevents the progressive acquisition of ventricular cardiomyocyte (VC) and pacemaker cardiomyocyte (PC) identities within distinct regions of the atrium. Transcriptomic analysis of flow-sorted atrial cardiomyocytes (ACs) from *nr2f1a* mutant zebrafish embryos showed increased VC marker gene expression and altered expression of core PC regulatory genes, including decreased expression of *nkx2.5*, a critical repressor of PC differentiation. At the arterial (outflow) pole of the atrium in *nr2f1a* mutants, cardiomyocytes resolve to VC identity within the expanded atrioventricular canal. However, at the venous (inflow) pole of the atrium, there is a progressive wave of AC transdifferentiation into PCs across the atrium toward the arterial pole. Restoring Nkx2.5 is sufficient to repress PC marker identity in *nr2f1a* mutant atria and analysis of chromatin accessibility identified an Nr2f1a-dependent *nkx2.5* enhancer expressed in the atrial myocardium directly adjacent to PCs. CRISPR/Cas9-mediated deletion of the putative *nkx2.5* enhancer leads to a loss of Nkx2.5-expressing ACs and expansion of a PC reporter, supporting that Nr2f1a limits PC differentiation within venous ACs via maintaining *nkx2.5* expression. The Nr2f-dependent maintenance of AC identity within discrete atrial compartments may provide insights into the molecular etiology of concurrent structural congenital heart defects and associated arrhythmias.

## Editor's evaluation

Addressing the role of NR2F transcription factors in the fish heart, this important work provides novel insight into atrial chamber patterning. Solid evidence supports the identification of a new mechanism of pacemaker cell restriction, involving nkx2.5 maintenance in the atrium by nr2f1a. Whether nr2f1 also restricts ventricular fate as shown in the mouse model will require more investigation. This manuscript will be of interest to developmental biologists and pediatric cardiologists.

## Introduction

The vertebrate heart relies on the coordination of multiple specialized cardiomyocyte populations. Cardiomyocytes of the atrium, ventricle, atrioventricular canal (AVC), and pacemaker are each

characterized by distinct mechanical, morphological, and electrophysiological properties (*Bootman et al., 2006*; *Brandenburg et al., 2016*; *Christoffels et al., 2010*; *Keith and Flack, 1907*; *Ng et al., 2010*; *Smyrnias et al., 2010*). Pacemaker cardiomyocytes (PCs) of the sinoatrial node (SAN), which is located at the venous pole of the single atrial chamber in zebrafish and the base of the right atrial chamber in mammals, initiate the synchronized wave of contraction that passes through the working atrial cardiomyocytes (ACs), slows at the AVC, and then rapidly activates ventricular cardiomyocytes (VCs) (*Arrenberg et al., 2010*; *Burkhard et al., 2017*; *Christoffels et al., 2010*; *Moorman and Christoffels, 2003*). Specific gene regulatory programs confer the differentiation and maintenance of these different cardiomyocyte populations within the heart during development (*Barth et al., 2005*; *Ng et al., 2010*; *Pradhan et al., 2017*; *Tabibiazar et al., 2003*; *Targoff et al., 2013*; *van Weerd and Christoffels, 2016*; *Xin et al., 2007*). Importantly, mutations in genes associated with promoting cardiomyocyte differentiation as well as the maintenance of identity, such as the transcription factors (TFs) *NKX2.5*, *TBX5*, and *NR2F2*, are associated with human congenital heart defects (CHDs), the most common type of congenital malformations found in newborn children (*Al Turki et al., 2014*; *Benjamin et al., 2018*; *Benson et al., 1999*; *Cheng et al., 2011*; *Hoffman and Kaplan, 2002*; *Loffredo, 2000*; *Nakamura et al., 2011*; *Schott et al., 1998*; *van der Linde et al., 2011*). Moreover, structural CHDs are often accompanied by arrhythmias that cause additional complications, morbidity, and death (*Bruneau et al., 1999*; *Ellesøe et al., 2016*; *Williams and Perry, 2018*). In adults, variants in *NKX2.5* and *TBX5* are also associated with isolated arrhythmias, such as atrial fibrillation, and improper cardiomyocyte gene expression (*Benson et al., 1999*; *Bruneau et al., 1999*; *Guo et al., 2016*; *Jhaveri et al., 2018*; *Ma et al., 2016*; *Nakashima et al., 2014*; *Yu et al., 2014*). Thus, the acquisition and maintenance of an appropriate number and identity of each cardiomyocyte population within the developing heart is vital for its normal function throughout life.

The Nr2f (Coup-tf) family of nuclear hormone receptor TFs are conserved regulators of atrial development. In humans, mutations in *NR2F2* have been associated with multiple types of CHDs, most commonly atrioventricular septal defects (AVSDs) (*Al Turki et al., 2014*; *Nakamura et al., 2011*; *Qiao et al., 2018*; *Upadia et al., 2018*). Consistent with a role in AC differentiation, NR2F1 and NR2F2 are expressed in ACs of the developing hearts of mice and humans (*Al Turki et al., 2014*; *Li et al., 2016*). In vitro studies have shown that both NR2F1 and NR2F2 promote AC differentiation in human embryonic stem cell (ESC)-derived cardiomyocytes (*Churko et al., 2018*; *Devalla et al., 2015*). Global *Nr2f2* knockout (KO) mice are embryonic lethal with prominent heart defects, including severely hypomorphic atria that lack septa, which is associated with reduced AC differentiation (*Pereira et al., 1999*). However, at subsequent stages, Nr2f2 appears to maintain atrial identity by directly repressing TFs including *Irx4* and *Hey2*, which promote and maintain the differentiation of VCs. Hence, cardiac-specific conditional KO of *Nr2f2* in mice results in ventricularized atria, while its ectopic expression in VCs is sufficient to promote AC identity (*Wu et al., 2013*). Although Nr2f2 appears to be the primary regulator of AC differentiation and identity in vivo in mammals, we have found that zebrafish Nr2f1a is the functional homolog of mammalian Nr2f2 with respect to heart development (*Dohn et al., 2019*; *Duong et al., 2018*). *Nr2f1a* mutant zebrafish develop smaller atria due to reduced AC differentiation at the venous pole and fail to restrict the size of the AVC, which results in an expansion of AVC markers into the atrium (*Duong et al., 2018*). Despite the established requirements for Nr2f TFs in promoting and maintaining atrial differentiation, the mechanisms by which Nr2f TFs function within vertebrate ACs are still not well understood.

Interactions between different regulatory programs determine the number and proportions of cardiomyocytes within the ventricle, AVC, atrium, and pacemaker that initially differentiate and are continuously required to reinforce their identity, which maintains compartmental boundaries within the heart that allow it to function properly. At the venous pole of the heart, a conserved regulatory network of TFs including Isl1, Shox/Shox2, and Tbx3 drives PC differentiation and concurrently represses AC identity. Loss of these TFs results in ectopic expression of working AC markers within the SAN and, in turn, a hypoplastic SAN; conversely, ectopic Tbx3 in working murine cardiomyocytes is sufficient to confer PC identity, and overexpression of Shox2 in murine ESCs during differentiation leads to an upregulation of PC genes (*Espinoza-Lewis et al., 2009*; *Espinoza-Lewis et al., 2011*; *Hoogaars et al., 2007*; *Ionta et al., 2015*; *Liang et al., 2015*; *Liu et al., 2011*; *Tessadori et al., 2012*; *van Weerd and Christoffels, 2016*). Nkx2.5 performs multiple functions within cardiomyocytes during heart development, consistent with pleiotropic structural and conduction defects found in patients

with *NKX2.5* mutations (*Benson et al., 1999*; *Ellesøe et al., 2016*; *Elliott et al., 2003*; *Jhaveri et al., 2018*; *McElhinney et al., 2003*; *Schott et al., 1998*; *Xie et al., 2013*; *Yu et al., 2014*). At the venous pole of the heart, it is a critical repressor of the PC differentiation regulatory network within working ACs, as loss of Nkx2.5 results in an expansion of PC marker identity within the atrium of mice and zebrafish (*Colombo et al., 2018*; *Espinoza-Lewis et al., 2011*; *Nakashima et al., 2014*). Within VCs at the arterial pole of the heart, Nkx2.5, as well as the TFs Irx4 and Hey2, maintain ventricular identity and repress the atrial gene program. Loss of these genes in mice and zebrafish results in ectopic expression of AC markers within the ventricle (*Bao et al., 1999*; *Bruneau et al., 2000*; *Koibuchi and Chin, 2007*; *Pradhan et al., 2017*; *Targoff et al., 2013*). Moreover, in embryonic zebrafish hearts, AC identity is repressed in VCs by FGF signaling, which functions upstream of Nkx2.5 and by inhibition of BMP (*de Pater et al., 2012*; *Pradhan et al., 2017*). However, in mice, FGF signaling is also necessary for proper ventricular conduction through regulation of connexin 43 (*Sakurai et al., 2013*). While the repression of AC identity is central to normal development of the ventricle and SAN, we still lack an understanding of mechanisms that maintain AC identity and plasticity within the embryonic vertebrate heart.

Here, we demonstrate that Nr2f1a is required to inhibit the progressive acquisition of VC and PC identity within different regions of the embryonic zebrafish atrium. At the arterial (outflow) pole of the atrium, Nr2f1a represses VC identity, similar to what has been shown in conditional *Nr2f2* KO mice. However, we find it is only sensitized cardiomyocytes within the expanded AVC of *nr2f1a* mutants, which co-express AC and VC differentiation markers, that progressively resolve to express only VC markers. Importantly, at the venous (inflow) pole of the atrium, we identify a requirement for Nr2f1a in repressing PC identity. While the initial PC population differentiates normally in *nr2f1a* mutants despite reduced AC differentiation, there is a subsequent expansion of PC markers and complementary loss of Nkx2.5 from the venous pole into the atrium. Electrophysiological studies demonstrate that the transdifferentiated ACs in *nr2f1a* mutants are adopting central conduction system identity. Genetic epistasis using a heat-shock inducible *nkx2.5* transgene showed that overexpressing Nkx2.5 is sufficient to inhibit the transdifferentiation of ACs that occurs in *nr2f1a* mutant hearts, indicating that within ACs Nr2f1a functions upstream of Nkx2.5 in the conserved genetic hierarchy that limits PC differentiation. Chromatin accessibility integrated with our transcriptomic analysis of sorted ACs identified a putative Nr2f1a-dependent *nkx2.5* enhancer that is expressed in a specific subset of venous ACs bordering the SAN. Deletion of this enhancer results in a loss of Nkx2.5 expression and an expansion of a PC reporter within ACs at the venous pole of the atrium. Overall, our results demonstrate that maintenance of Nr2f expression is critical for normal vertebrate heart development through sustaining atrial identity in different AC subpopulations, which may provide insights into the etiology of concomitant congenital structural cardiac and conduction defects found in humans.

## Results

### Cardiomyocytes within the AVC progressively resolve to VC identity in *nr2f1a* mutants

Although previous studies have indicated that Nr2f TFs are required to promote atrial differentiation (*Duong et al., 2018*; *Pereira et al., 1999*; *Wu et al., 2013*), we still do not fully understand the effects of Nr2f loss on AC identity. To investigate the consequences of Nr2f loss within the atrium, we performed RNA-seq on flow-sorted cardiomyocytes expressing the AC differentiation marker *atrial myosin heavy chain* (*amhc*; also called *myh6*) from wild-type (WT) and *nr2f1a* mutant embryos. Embryos carrying the *Tg(amhc:EGFP)* transgene (*Zhang et al., 2013*) were dissociated at 48 hr post-fertilization (hpf) and fluorescence-activated cell sorting (FACS) was performed to isolate *amhc:*EGFP+ cardiomyocytes (*Figure 1A*). The resulting RNA-seq dataset revealed 2077 upregulated and 1297 downregulated genes with >2-fold change in the *nr2f1a* mutant *amhc:*EGFP+ cardiomyocytes relative to WT (*Supplementary file 1*). Genes associated with AC differentiation, such as *amhc* itself, were decreased in the *amhc:*EGFP+ cardiomyocytes captured from *nr2f1a* mutant embryos relative to the WT embryos, confirming the success of this approach (*Figure 1B*; *Supplementary file 1*). However, we also observed two additional trends in the *nr2f1a* mutant *amhc:*EGFP+ cardiomyocytes: an increase in the expression of genes associated with VC and outflow tract identity of the heart, such as the *irx4* and *isl2* paralogs (*Bruneau et al., 2000*; *Bruneau et al., 2001*; *Targoff et al., 2013*; *Witzel et al.,*

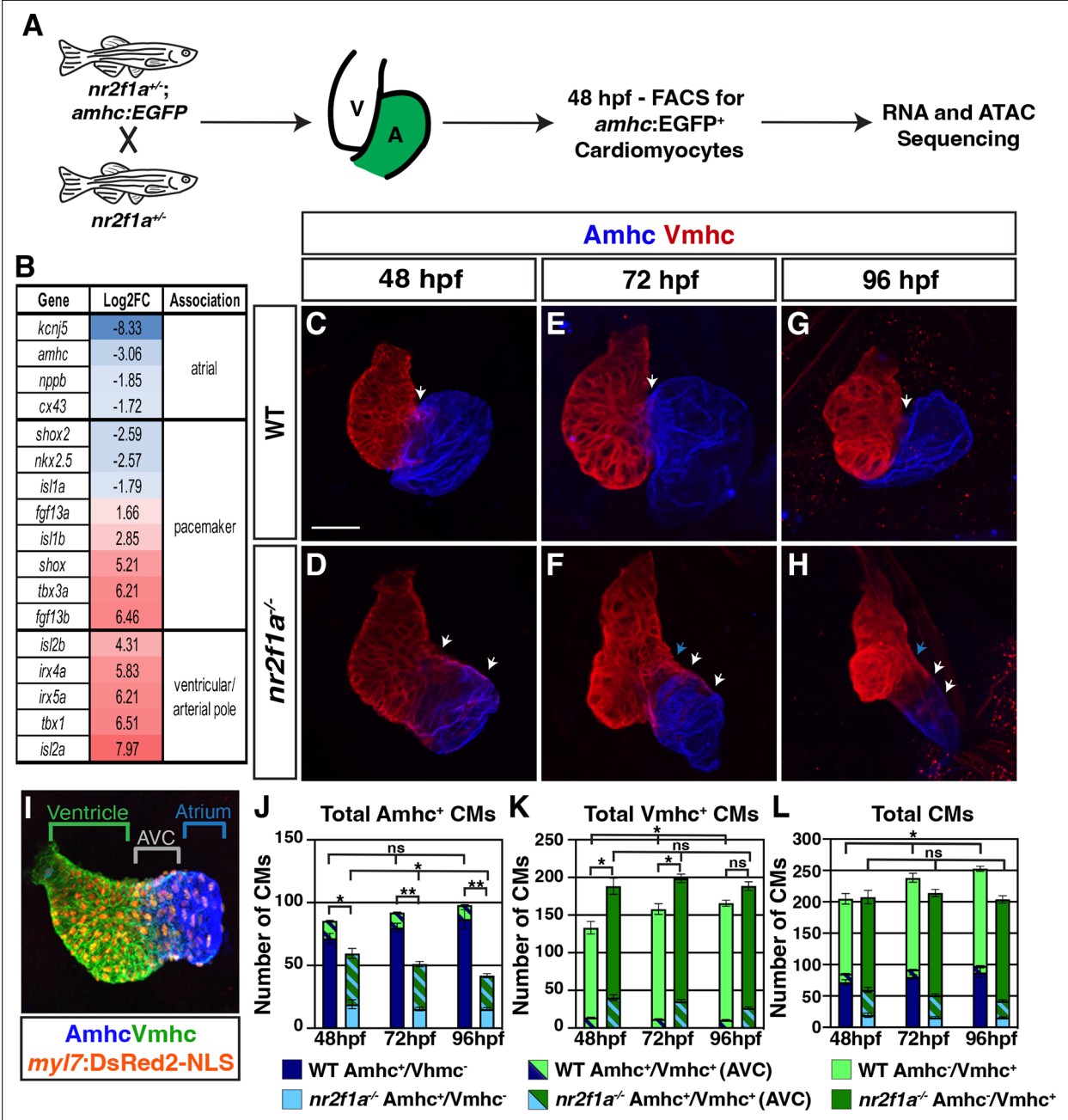

**Figure 1.** The atrioventricular canal (AVC) resolves to Vmhc-expressing cardiomyocytes in *nr2f1a* mutant hearts. (**A**) Schematic for isolation of atrial cardiomyocytes (ACs) using *Tg(amhc:EGFP)* transgene for RNA-seq and assay for transpose-accessible chromatin sequencing (ATAC-seq) at 48 hr post-fertilization (hpf). (**B**) Differential expression of genes associated with ventricular cardiomyocyte (VC)/arterial pole, AC, and pacemaker cardiomyocyte (PC) differentiation in *nr2f1a* mutants compared to wild-type (WT). (**C–H**) IHC for Amhc (blue) and Vmhc (red) in *nr2f1a* and WT hearts at 48, 72, and 96 hpf. White arrows indicate region of overlapping Amhc+ and Vmhc+ cardiomyocytes in the AVC. Blue arrows indicate overt arterial position of the AVC of *nr2f1a* mutant atria. Number of embryos examined - 48 hpf: WT (n=3), *nr2f1a⁻/⁻* (n=4); 72 hpf: WT (n=4), *nr2f1a⁻/⁻* (n=6); 96 hpf: WT (n=7), *nr2f1a⁻/⁻* (n=10). (**I**) Schematic for cell quantification: Amhc+/Vmhc− (blue) cardiomyocytes (CMs) mark the atria; Amhc+/Vmhc+ (green/blue) cardiomyocytes mark the AVC; Amhc−/Vmhc+ (green) cardiomyocytes mark the ventricles; DsRed2-NLS (pan-cardiac) marks the cardiomyocyte nuclei. (**J–L**) Quantification of Amhc+/Vmhc− cardiomyocytes, Amhc+/Vmhc+ (AVC) cardiomyocytes, and Amhc−/Vmhc+ cardiomyocytes. Error bars indicate s.e.m. 48 hpf: WT (n=4), *nr2f1a⁻/⁻* (n=5); 72 hpf: WT (n=5), *nr2f1a⁻/⁻* (n=8); 96 hpf: WT (n=5), *nr2f1a⁻/⁻* (n=12). Scale bar indicates 50 μm. Differences between WT and *nr2f1a⁻/⁻* were analyzed using ANOVA with multiple comparisons. *p=0.05–0.001, **p<0.001.

The online version of this article includes the following figure supplement(s) for figure 1:

*Figure 1 continued on next page*

*Figure 1 continued*

**Figure supplement 1.** Individual channels showing the resolution of atrioventricular canal (AVC) cardiomyocytes to ventricular cardiomyocyte (VC) identity in *nr2f1a* mutants.

**Figure supplement 2.** Quantification of cardiomyocytes in *nr2f1a* mutants.

**Figure supplement 3.** *Nr2f1a* mutant hearts do not have increased cardiomyocyte death.

**Figure supplement 4.** Cardiomyocytes within the *nr2f1a* mutant atria do not have increased proliferation.

*2017*; *Wu et al., 2013*); and misexpression of core genes associated with PC differentiation, with genes shown to promote PC differentiation, such as *tbx3a* and *shox*, being increased (*Hoogaars et al., 2007*; *Kenyon et al., 2011*; *Liu et al., 2011*; *Ribeiro et al., 2007*; *Sawada et al., 2015*), while *nkx2.5*, a critical repressor of PC differentiation, was decreased (*Colombo et al., 2018*; *Espinoza-Lewis et al., 2011*; *Nakashima et al., 2014*; *Figure 1B*). Thus, transcriptional analysis implies that *nr2f1a* mutant atria have increased expression of genes associated with promoting both VC and PC differentiation.

Conditional KO of *Nr2f2* in murine ACs results in their acquisition of VC identity (*Wu et al., 2013*). Furthermore, while our transcriptomic analysis here and previous analysis of the ventricular differentiation marker *ventricular myosin heavy chain* (*vmhc*; also called *myh7*) (*Duong et al., 2018*) also suggests there may be an expansion of VC/arterial pole identity into the atrium of *nr2f1a* mutants, it was still not clear if a fate switch of ACs to VCs occurred within *nr2f1a* mutant atria at 48 hpf comparable to what was reported in the conditional *Nr2f2* KO mice (*Wu et al., 2013*). To examine cardiomyocyte identities more closely within *nr2f1a* mutant hearts, we performed immunohistochemistry (IHC) for the cardiomyocyte differentiation markers Amhc and Vmhc (*Figure 1C–H*) and quantified cardiomyocyte number by using the pan-cardiac *Tg(myl7:DsRed2-NLS)* transgene (*Figure 1I*; *Mably et al., 2003*). Although we found a significant overlap of Amhc and Vmhc in the heart tubes at 30 hpf, which was indistinguishable between WT and *nr2f1a* mutant hearts (*Figure 1—figure supplement 1A–A″, E–E″*), the number of Amhc⁺/Vmhc⁺ cardiomyocytes, which we classify as cardiomyocytes of the AVC, becomes significantly restricted by 48 hpf in the hearts of WT embryos (*Figure 1C and J–L*; *Figure 1—figure supplement 1B–B″*). Subsequently, from 48 to 96 hpf there remained a relatively constant and small number of Amhc⁺/Vmhc⁺ cardiomyocytes in the hearts of WT embryos (*Figure 1C, E, G and J–L*; *Figure 1—figure supplement 1B–D″*; *Figure 1—figure supplement 2B*). However, by 48 hpf in *nr2f1a* mutant hearts, there were still a significant number of cardiomyocytes co-expressing Amhc and Vmhc (*Figure 1D and J–L*; *Figure 1—figure supplement 1F–F″*; *Figure 1—figure supplement 2B*), consistent with their expanded AVC that we reported previously (*Duong et al., 2018*). Interestingly, the number of Amhc⁺/Vmhc⁺ cardiomyocytes within this region of *nr2f1a* mutant hearts was progressively reduced through 96 hpf (*Figure 1F, H and J–L*; *Figure 1—figure supplement 1G–G″, H-H″*; *Figure 1—figure supplement 2B*). While Amhc⁺/Vmhc⁺ cardiomyocytes remained constant over these stages, there was a modest increase of Amhc⁺/Vmhc⁻ cardiomyocytes in the atria of WT hearts. However, total Amhc⁺ cardiomyocytes (Amhc⁺/Vmhc⁻ and Amhc⁺/Vmhc⁺) were found to diminish from 48 to 96 hpf in the *nr2f1a* mutant hearts, which was primarily due to the decrease in Amhc⁺/Vmhc⁺ cardiomyocytes as the number of Amhc⁺/Vmhc⁻ cardiomyocytes, while dramatically reduced compared to WT hearts, did not change significantly over these stages (*Figure 1J*; *Figure 1—figure supplement 1F–H″* and *Figure 1—figure supplement 2A and B*). Conversely, there were more total Vmhc⁺ (Amhc⁻/Vmhc⁺ and Amhc⁺/Vmhc⁺) cardiomyocytes in *nr2f1a* mutant hearts compared to WT hearts across all timepoints, which primarily reflects the expansion of Vmhc expression within the enlarged AVC of the *nr2f1a* mutant hearts (*Figure 1K*; *Figure 1—figure supplement 1F–H″* and *Figure 1—figure supplement 2B and C*). Furthermore, the number of Vmhc⁺-only (Amhc⁻/Vmhc⁺) cardiomyocytes progressively increased from 48 to 96 hpf in *nr2f1a* mutant hearts, complementing the loss of Amhc⁺/Vmhc⁺ cardiomyocytes within the AVC (*Figure 1K*; *Figure 1—figure supplement 1F–H″* and *Figure 1—figure supplement 2C and D*). Interestingly, the total number of cardiomyocytes in *nr2f1a* mutant hearts remained constant from 48 to 96 hpf, stages that showed a progressive increase in the total number of cardiomyocytes in WT hearts (*Figure 1L*), implying that Nr2f1a loss within the atrium affects overall heart growth. Cell death, as assayed with IHC for active Caspase 3, and proliferation, as assayed with IHC for phosphohistone H3 (pHH3), were not affected at these stages in *nr2f1a* mutant hearts (*Figure 1—figure supplement 3A–D*; *Figure 1—figure supplement*

*4A–F*). Similarly, EdU pulses at 48 or 72 hpf showed that Amhc$^+$ cardiomyocytes in *nr2f1a* mutants hearts also do not have reduced proliferation (*Figure 1—figure supplement 4G–M*). However, Amhc$^-$ cardiomyocytes, which are predominantly VCs, did show reduced proliferation (*Figure 1—figure supplement 4G–M*), suggesting the lack of increase in heart size in *nr2f1a* mutants primarily reflects a lack of ventricular growth. Thus, our data indicate that changes in the different cardiomyocyte populations of *nr2f1a* mutants hearts are not due to increased cell death or proliferation. However, the lack of observed heart growth is potentially due to Nr2f1a loss cell non-autonomously affecting proliferation of VCs, which will be investigated in the future. Collectively, these results show that the expansion and acquisition of VC identity within *nr2f1a* mutant hearts is regionally restricted, with cardiomyocytes comprising the expanded AVC that initially expresses both AC and VC differentiation markers resolving to VC identity.

## PC identity progressively expands throughout *nr2f1a* mutant atria

Our results show that in the absence of Nr2f TFs not all Amhc$^+$ cardiomyocytes within the atrium acquire VC identity. Moreover, our transcriptomic data indicated there was also misexpression of core regulatory genes that control PC differentiation in the *nr2f1a* mutant Amhc$^+$ cardiomyocytes (*Figure 1A and B*). Orthologs of genes associated with promoting PC differentiation and function, including *tbx3a*, *shox*, and *fgf13a* (*Burkhard and Bakkers, 2018*; *Colombo et al., 2018*; *Espinoza-Lewis et al., 2009*; *Hoogaars et al., 2007*; *Kenyon et al., 2011*; *Liu et al., 2011*; *Poon et al., 2016*; *Ribeiro et al., 2007*; *Sawada et al., 2015*; *Wang et al., 2011*; *Wang et al., 2017*; *Yang et al., 2016*), were predominantly increased, while *nkx2.5*, a critical repressor of PC identity in mice and zebrafish (*Colombo et al., 2018*; *Hoogaars et al., 2007*; *Nakashima et al., 2014*), was decreased. To determine if there are effects on the PCs in *nr2f1a* mutants, we used the *SqET33-mi59B* enhancer trap line, which for clarity we refer to as *Et(fgf13a:EGFP)*, as it reports *fgf13a* expression and is expressed in PCs (*Poon et al., 2016*). At 48 hpf, we found no overt difference in *Et(fgf13a:EGFP)* expression between the WT and *nr2f1a* mutant embryos (*Figure 2A and B*; *Figure 2—figure supplement 1A–A‴, D–D‴*), despite the smaller atria in the *nr2f1a* mutants. However, while *Et(fgf13a:EGFP)* expression is normally restricted to the venous pole in WT hearts, we found a progressive expansion of *Et(fgf13a:EGFP)* expression from the venous pole throughout the Amhc$^+$ cardiomyocytes in the atrium of the *nr2f1a* mutant hearts through 96 hpf (*Figure 2C–F*; *Figure 2—figure supplement 1B–C‴, E–F‴*). Quantification of Amhc$^+$ cardiomyocytes co-expressing the *Et(fgf13a:EGFP)* transgene via incorporation of the *Tg(myl7:DsRed2-NLS)* transgene confirmed the concurrent increase in *fgf13a*:EGFP$^+$/Amhc$^+$ cardiomyocytes and decrease in *fgf13a*:EGFP$^-$/Amhc$^+$ cardiomyocytes within the *nr2f1a* mutant atria (*Figure 2G and H*), even as the total number of Amhc$^+$ cardiomyocytes decreases (*Figure 2I*). Equivalent results were found with IHC for the SAN marker Isl1 (*Liang et al., 2015*; *Sun et al., 2007*; *Tessadori et al., 2012*; *Figure 2—figure supplement 2*). Additionally, real-time quantitative PCR (RT-qPCR) on flow-sorted cardiomyocytes from 96 hpf embryos using the pan-cardiac *Tg(myl7:EGFP)* transgene (*Huang et al., 2003*) showed an increase in PC differentiation marker expression, as well as *vmhc* expression, and decrease in *amhc* at these stages in *nr2f1a* mutant hearts (*Figure 2J and K*), reflecting the progressive expansion observed with expression of *Et(fgf13a:EGFP)* and Isl1 with IHC analysis.

Because the total number of cardiomyocytes in the hearts of the *nr2f1a* mutants does not change over the stages examined (*Figure 1L*), the aforementioned analysis of cardiomyocyte death and proliferation (*Figure 1—figure supplement 3* and *Figure 1—figure supplement 4*) would suggest that the arterial-directed expansion of *Et(fgf13a:EGFP)* expression in Amhc$^+$ cardiomyocytes reflects an acquisition of PC identity in these cardiomyocytes and is not due to the incorporation of newly differentiating cardiomyocytes at the venous pole of the heart. We sought to further test this hypothesis via lineage tracing of cardiomyocytes using a temporal differentiation assay with the transgene *Tg(myl7:NLS-KikGR)*, which expresses the photoconvertible NLS-KikGR in all cardiomyocytes (*Lazic and Scott, 2011*). The cardiomyocyte-specific NLS-KikGR was photoconverted from green to red in the hearts of embryos also carrying the *Et(fgf13a:EGFP)* transgene at 72 hpf. The hearts were then analyzed at 96 hpf for the presence of red and green *fgf13a*:EGFP$^+$ cardiomyocytes (*Figure 2—figure supplement 3A*). Importantly, the green, nuclear KikGR was visible within the cardiomyocytes even with expression of the cytosolic *fgf13a*:EGFP. We found that in both WT and *nr2f1a* mutant hearts all *fgf13a*:EGFP$^+$ cardiomyocytes within the atria co-expressed both photoconverted (red) NLS-KikGR and newly expressed (green) NLS-KikGR, including those in the expanded *fgf13a*:EGFP$^+$ expression

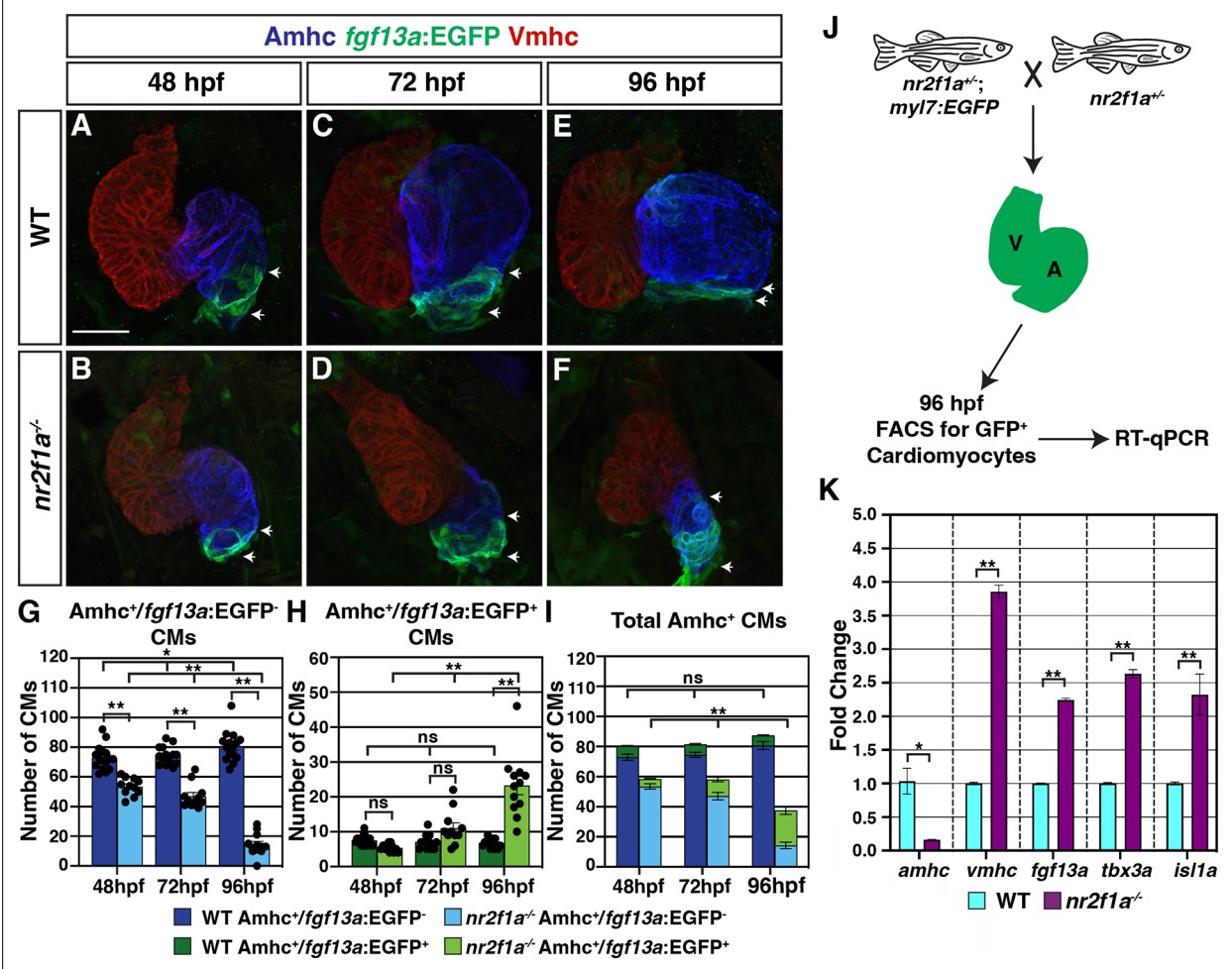

**Figure 2.** Pacemaker cardiomyocyte (PC) identity expands from the venous pole across the atrium in *nr2f1a* mutant hearts. (**A–F**) IHC for Amhc (blue), Vmhc (red), and *fgf13a*:EGFP (green). White arrows indicate boundaries of *Et(fgf13a:EGFP)* expression within the atrium. Number of embryos examined - 48 hr post-fertilization (hpf): wild-type (WT) (n=10), *nr2f1a⁻/⁻* (n=12); 72 hpf: WT (n=10), *nr2f1a⁻/⁻* (n=10); 96 hpf: WT (n=7), *nr2f1a⁻/⁻* (n=10). (**G–I**) Quantification of Amhc⁺/*fgf13a*:EGFP⁻, Amhc⁺/*fgf13a*:EGFP⁺, and total Amhc⁺ cardiomyocytes (CMs) using the *Tg(myl7:DsRed2-NLS)* transgene within the hearts of WT and *nr2f1a* mutants. Error bars indicate s.e.m.; 48 hpf: WT (n=16), *nr2f1a⁻/⁻* (n=11); 72 hpf: WT (n=14), *nr2f1a⁻/⁻* (n=11); 96 hpf: WT (n=15), *nr2f1a⁻/⁻* (n=12). (**J**) Schematic for the isolation of cardiomyocytes at 96 hpf using the *Tg(myl7:EGFP)* transgene. (**K**) Fold change of marker genes relative to *β-actin* from real-time quantitative PCR (RT-qPCR) on isolated cardiomyocytes of WT and *nr2f1a* mutants at 96 hpf. Scale bar indicates 50 μm. Differences between WT and *nr2f1a⁻/⁻* were analyzed using ANOVA with multiple comparisons. *p=0.05–0.001, **p<0.001.

The online version of this article includes the following figure supplement(s) for figure 2:

**Figure supplement 1.** Individual channels showing that pacemaker cardiomyocyte (PC) identity expands from the venous pole across the atrium in *nr2f1a* mutant hearts.

**Figure supplement 2.** Isl1 expression expands throughout the atrium in *nr2f1a* mutant embryos.

**Figure supplement 3.** Newly differentiating cardiomyocytes are not added to the atrium in *nr2f1a* mutant hearts.

domain and venous pole of *nr2f1a* mutant hearts (*Figure 2—figure supplement 3B–E''*). The absence of green-only, newly differentiated NLS-KikGR⁺ cardiomyocytes in the *nr2f1a* mutant atria at 96 hpf indicates that all of the *fgf13a*:EGFP⁺ cardiomyocytes were present in the atrium at 72 hpf. Thus, our data support that venous Amhc⁺ ACs are progressively transdifferentiating into PCs in the absence of Nr2f1a.

Next, we wanted to determine if the ACs that are acquiring PC marker gene expression in the *nr2f1a* mutants are functionally adopting PC characteristics. PCs have different electrophysiological properties than working cardiomyocytes, including slower depolarization and the ability to spontaneously generate action potentials (*Christoffels et al., 2010*; *Lakatta et al., 2010*; *Mesirca et al., 2015*; *Schram et al., 2002*). Moreover, given that the intrinsic automaticity of coupled cells is a function of

their total electrical capacitance (*Honjo et al., 1996*; *Joyner and van Capelle, 1986*; *Mangoni and Nargeot, 2008*; *Watanabe et al., 1995*), one might predict that if there are more ACs adopting PC or central conduction system-like characteristics then the heart rates of *nr2f1a* mutants may be slower than WT embryos. We assessed heart rate using high-speed time lapse imaging (*Figure 3A and B*; *Figure 3—videos 1–6*). At 48 hpf, there was no difference in heart rate between WT and *nr2f1a* mutant hearts (*Figure 3C*). However, as the heart rate of WT embryos increased through 96 hpf, the heart rate of *nr2f1a* mutant hearts failed to increase (*Figure 3C*). To examine the electrophysiological properties of the ACs that are acquiring PC marker expression in *nr2f1a* mutant hearts, we analyzed myocardial electrophysiology using optical voltage mapping (*Colombo et al., 2018*; *Mosimann et al., 2015*; *Panáková et al., 2010*). The intrinsic bradycardia we observed appeared to be related to effects on the duration and extent of repolarization with evidence of modest hyperpolarization in both heterozygous and homozygous *nr2f1a* mutant hearts, as well as substantial prolongation of repolarization in the homozygous *nr2f1a* mutant hearts, as conservatively assessed based on the action potential duration at 20% repolarization (APD20) (*Figure 3D*). Measuring the APD20 within the center of the atria (*Figure 3A*), which acquires PC marker expression in the *nr2f1a* mutants based on the IHC analysis, showed that even at 48 hpf the ACs in *nr2f1a* mutant hearts depolarize more slowly compared to WT and heterozygous *nr2f1a* atria (*Figure 3E*; *Figure 3—figure supplement 1A*). These effects were not associated with any changes in Phase 4 depolarization (data not shown). There was evidence of significantly slower conduction across the *nr2f1a* mutant atria by 96 hpf (*Figure 3F–L*; *Figure 3—figure supplement 1B*), but with no effect on Vmax (*Figure 3—figure supplement 1C*), consistent with less chamber-like and more central conduction system-like intercellular coupling characteristics. Remarkably, there was no evidence of the emergence of the AVC slow conduction zone in the *nr2f1a* mutant hearts that is characteristic of the AV junction and the resultant sequential chamber contraction that is the hallmark of the vertebrate heart (*Figure 3G–L*), supporting that the cardiomyocytes in the AVC of *nr2f1a* mutants also have acquired intermediate physiological properties with the VCs. Collectively, our data support that more venous ACs in *nr2f1a* mutant hearts are adopting a functional identity that exhibits delayed repolarization and reduced intercellular coupling, closer to the physiology of the cardiomyocytes of the central conduction system and paralleling the PC-associated gene expression changes we observe within the *nr2f1a* mutant atria.

## Nr2f1a represses SAN identity upstream of Nkx2.5

We next wanted to understand how Nr2f1a may be regulating the PC gene regulatory network within the atrium. We prioritized examining Nkx2.5 in *nr2f1a* mutant hearts, as Nkx2.5 loss in mice and zebrafish results in an expansion of PCs within the atrium (*Colombo et al., 2018*; *Espinoza-Lewis et al., 2011*; *Nakashima et al., 2014*) and its expression was reduced in the bulk RNA-seq data of *nr2f1a* mutant *amhc*:EGFP[+] cardiomyocytes (*Figure 1B*). IHC for Nkx2.5 and either the *Et(fgf13a:EGFP)* reporter or Isl1 both showed that in WT hearts at 48 through 96 hpf Nkx2.5 is expressed throughout the atrial myocardium but is predominantly excluded from the PCs at the venous pole (*Figure 4A–C"*; *Figure 4—figure supplement 1*), consistent with what has been reported in zebrafish and mice (*Colombo et al., 2018*; *Mommersteeg et al., 2007*). At 48 hpf in *nr2f1a* mutants, Nkx2.5 shows a similar complementary pattern with the *Et(fgf13a:EGFP)* reporter despite the smaller atria (*Figure 4D*). As the *Et(fgf13a:EGFP)* reporter expands toward the AVC into the Amhc[+] ACs through 96 hpf in *nr2f1a* mutants, Nkx2.5 expression recedes from the venous pole and predominantly maintains the complementary expression pattern (*Figure 4E–F"*). Equivalent results were obtained using the *TgBAC(–36Nkx2.5:ZsYellow)* transgene (*Zhou et al., 2011*; *Figure 4—figure supplement 2*). Thus, a progressive loss of Nkx2.5 within ACs correlates with and abuts the expansion of PC identity in *nr2f1a* mutant atria.

To determine if restoring Nkx2.5 in *nr2f1a* mutants can repress the transdifferentiation of ACs to PCs, we used the *Tg(hsp70l:nkx2.5-EGFP)* transgene (*George et al., 2015*). *Nr2f1a*[+/-] fish that carry the *Tg(myl7:DsRed2-NLS)* and *Et(fgf13a:EGFP)* transgenes were crossed to *nr2f1a*[+/-] fish carrying the *Tg(hsp70l:nkx2.5-EGFP)* transgene (*Figure 5A*). The resulting embryos were sorted for the *Et(fgf13a:EGFP)* transgene and then heat-shocked at 20 hpf, as a single heat-shock at ~20 hpf using the *Tg(hsp70l:nkx2.5-EGFP)* transgene is sufficient to rescue *nkx2.5* mutants to adulthood (*George et al., 2015*). Following heat-shock, the embryos were sorted based on the presence of Nkx2.5-EGFP expression and allowed to develop until 48 and 96 hpf, at which point the Nkx2.5-EGFP is no longer

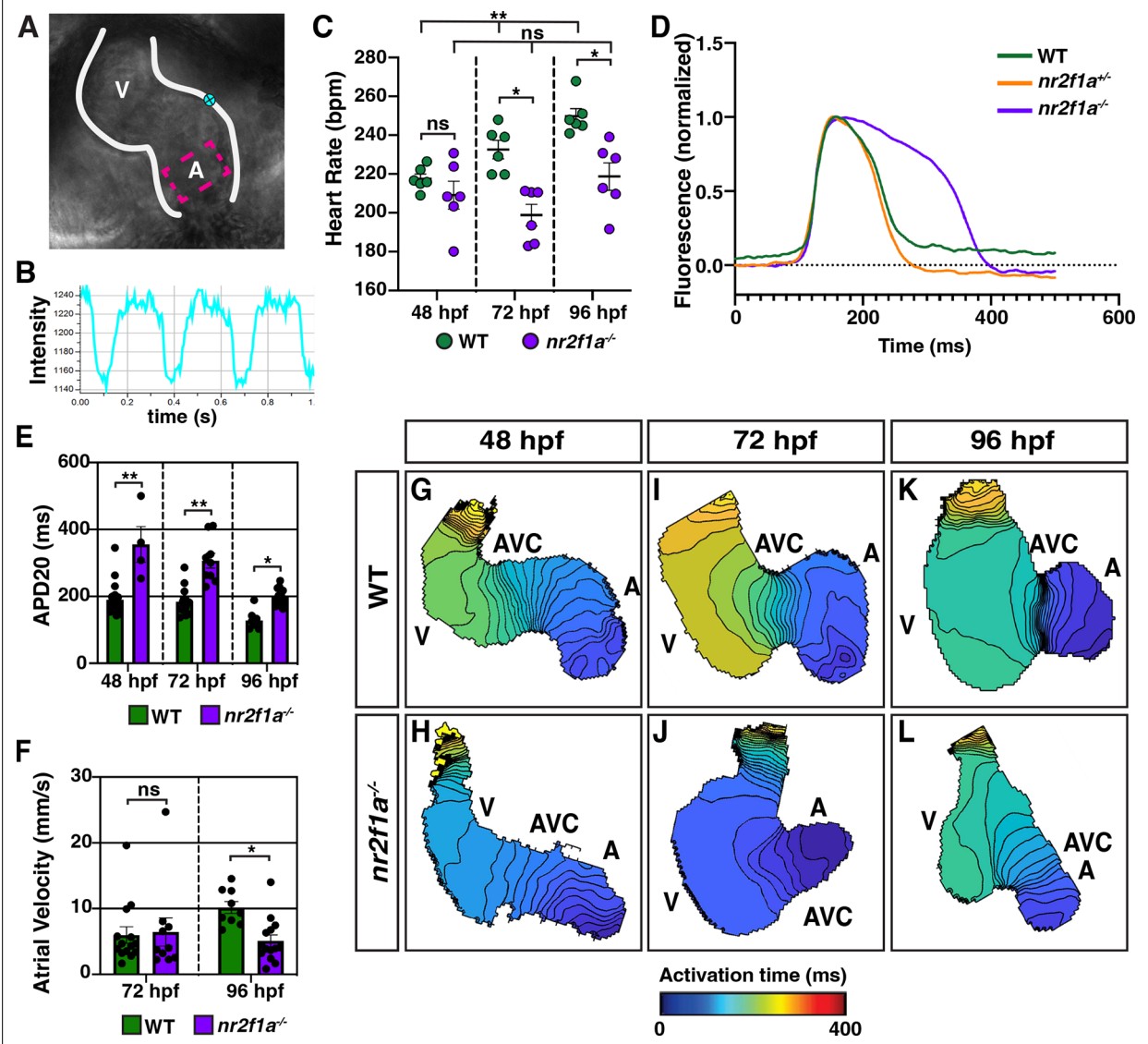

**Figure 3.** Atrial cardiomyocytes (ACs) in *nr2f1a* mutants function as pacemaker cardiomyocytes (PCs). (**A**) Schematic of point placements to measure heart rate (cyan) and action potential duration at 20% repolarization (APD20) (magenta) in a 48 hr post-fertilization (hpf) heart. (**B**) Representative kymograph used to analyze heart rate in wild-type (WT) and *nr2f1a* mutant embryos. (**C**) Quantification of heart rate in WT and *nr2f1a* mutant embryos; 48 hpf: WT (n=6), *nr2f1a⁻/⁻* (n=6); 72 hpf: WT (n=6), *nr2f1a⁻/⁻* (n=6); 96 hpf: WT (n=6), *nr2f1a⁻/⁻* (n=6). (**D**) Representative atrial action potentials of WT, *nr2f1a⁺/⁻*, and *nr2f1a⁻/⁻* embryos at 96 hpf. (**E**) APD20 in WT and *nr2f1a⁻/⁻* hearts; 48hpf: WT (n=21), *nr2f1a⁻/⁻* (n=4); 72 hpf: WT (n=13), *nr2f1a⁻/⁻* (n=9); 96 hpf: WT (n=9), *nr2f1a⁻/⁻* (n=14). (**F**) Atrial velocity in WT and *nr2f1a⁻/⁻* hearts; 72 hpf: WT (n=13), *nr2f1a⁻/⁻* (n=9); 96 hpf: WT (n=9), *nr2f1a⁻/⁻* (n=14). (**G–L**) Representative isochronal maps illustrating the positions of the depolarizing wave front in 5 ms intervals for WT and *nr2f1a⁻/⁻* embryos at 48, 72, and 96 hpf; 48 hpf: WT (n=20), *nr2f1a⁻/⁻* (n=6) 72 hpf: WT (n=12), *nr2f1a⁻/⁻* (n=7) 96 hpf: WT (n=11), *nr2f1a⁻/⁻* (n=14). Differences between WT and *nr2f1a⁻/⁻* analyzed using ANOVA with multiple comparisons. Error bars in all graphs indicate s.e.m. *p=0.05–0.001, **p<0.001.

The online version of this article includes the following video and figure supplement(s) for figure 3:

**Figure supplement 1.** Prolonged repolarization and decreased atrial conduction velocity in *nr2f1a* mutants compared to wild-type (WT) and heterozygous *nr2f1a* atria.

**Figure 3—video 1.** Heart from 48 hr post-fertilization (hpf) wild-type (WT) embryo.
https://elifesciences.org/articles/77408/figures#fig3video1

**Figure 3—video 2.** Heart from 48 hr post-fertilization (hpf) *nr2f1a* mutant embryo.
https://elifesciences.org/articles/77408/figures#fig3video2

**Figure 3—video 3.** Heart from 72 hr post-fertilization (hpf) wild-type (WT) embryo.
https://elifesciences.org/articles/77408/figures#fig3video3

*Figure 3 continued on next page*

visible, and the number of *fgf13a*:EGFP⁺ cardiomyocytes was quantified (*Figure 5A*). At 48 hpf, there was a modest decrease in the number of *fgf13a*:EGFP⁺ cardiomyocytes in Nkx2.5-EGFP⁺ WT hearts compared to the Nkx2.5-EGFP⁻ WT siblings (*Figure 5B, C, J and K*). However, Nkx2.5-EGFP⁺ *nr2f1a* mutants had a complete absence of *fgf13a*:EGFP⁺ cardiomyocytes at the venous pole (*Figure 5F, G, J and K*), suggesting that Nkx2.5 is sufficient to repress PC differentiation in *nr2f1a* mutant hearts and, further, that *nr2f1a* mutant ACs may be sensitized to increased Nkx2.5 expression. At 96 hpf, both Nkx2.5-EGFP⁺ WT and *nr2f1a* mutant hearts had an increased number of *fgf13a*:EGFP⁺ cardiomyocytes compared to those at 48 hpf, yet Nkx2.5-EGFP⁺ *nr2f1a* mutants still had significantly fewer *fgf13a*:EGFP⁺ cardiomyocytes than mutants lacking the transgene (*Figure 5D, E and H–K*). While we see a repression of the *fgf13a*:EGFP⁺ cardiomyocyte expansion at 48 hpf, we found that the overexpression of Nkx2.5 did not rescue the heart rate of *nr2f1a* mutants at 96 hpf when we observe the resumption of the PC reporter (*Figure 5—figure supplement 1*), suggesting that despite the inhibition of the reporter expansion, additional Nkx2.5-independent mechanisms downstream of Nr2f1a also contribute to the inhibition of PC identity. We found that heat-shock induction of Nkx2.5-EGFP at 40 hpf could also repress the expansion of *fgf13a*:EGFP⁺ cardiomyocytes in *nr2f1a* mutant atria (*Figure 5—figure supplement 2A–C, F, G, J and K*). However, the repression of *fgf13a*:EGFP⁺ cardiomyocytes within the mutant atria was not as effective or prolonged as the heat-shock induction of Nkx2.5-EGFP at 20 hpf (*Figure 5—figure supplement 2D, E and H–K*). Overall, these data suggest that Nr2f1a prevents the expansion of PC identity within the venous pole of the atrium at least in part through promoting or maintaining Nkx2.5 in ACs.

## A putative *nkx2.5* enhancer is expressed within the venous pole of the atrium

To identify cis-regulatory elements that may control *nkx2.5* expression and be affected by Nr2f1a loss, we assayed chromatin accessibility by performing assay for transpose-accessible chromatin sequencing (ATAC-seq) on flow-sorted *amhc*:EGFP⁺ cardiomyocytes from 48 hpf embryos, similar to what is described above (*Figure 1A*; *Figure 6—figure supplement 1*). Changes in called peaks of WT and *nr2f1a* mutant *amhc*:EGFP⁺ cardiomyocytes were scored relative to the nearest differentially expressed genes (*Supplementary file 1*). HOMER was used to determine enrichment for canonical Nr2f TF binding sites on the putative enhancer elements. One of the peaks that was revealed to close in *nr2f1a* mutant atria and contain an Nr2f binding site was ~55 kb upstream of the *nkx2.5* transcriptional start site (*Figure 6A*). Although this region did not show significant sequence conservation, enhancer conservation can vary between tissue types, with enhancers that drive expression in the heart often being weakly conserved (*Blow et al., 2010*). This putative enhancer region was placed into a reporter vector upstream of the basal *E1b* promoter to drive GFP expression (*Figure 6B*; *Birnbaum et al., 2012*), and transgenic lines were generated using Tol2 (*Kwan et al., 2007*). Remarkably, stable transgenic lines showed that this putative enhancer is expressed in a group of cardiomyocytes at the venous pole of the atrium (*Figure 6C*), which are directly adjacent to Isl1⁺ PCs (*Figure 6D and D'*). The *Tg(–55nkx2.5:EGFP)* transgene was then crossed into *nr2f1a* mutant allele carriers to determine if the expression of this putative *nkx2.5* enhancer is affected in *nr2f1a* mutants. We found the number of atria with *Tg(–55nkx2.5:EGFP)* expression was significantly decreased in *nr2f1a* mutants compared to WT, suggesting that Nr2f1a is required for enhancer expression or for those cardiomyocytes to develop (*Figure 6C, E and F*). Together, these data suggest that a putative Nr2f1a-dependent *nkx2.5* enhancer promotes expression in ACs adjacent to PCs.

To determine if the putative *nkx2.5* enhancer is required to promote Nkx2.5 expression and restrict PC identity within the venous pole of the atrium, we employed an established, highly efficient CRISPR/

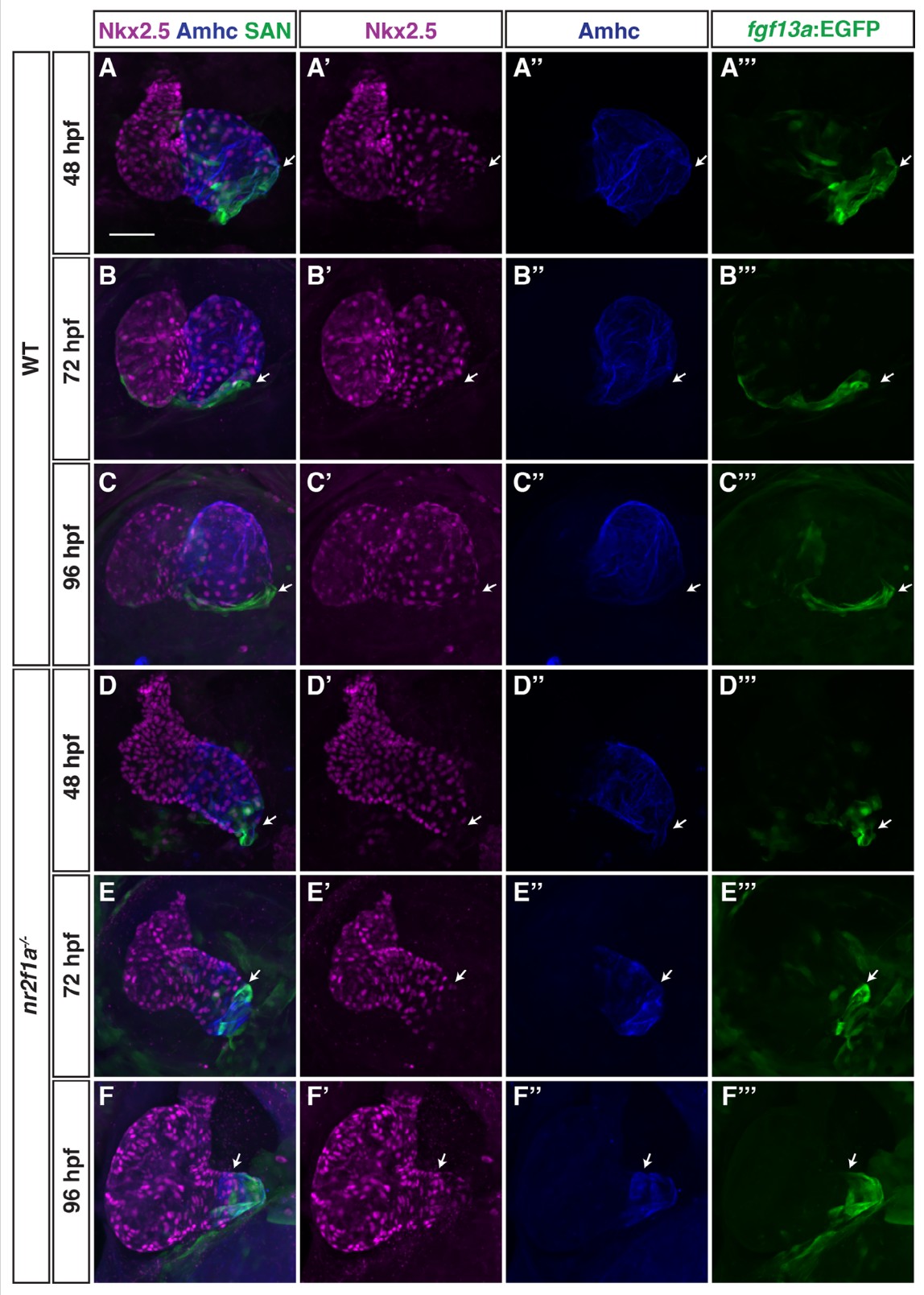

**Figure 4.** Nkx2.5 expression recedes from venous pole in *nr2f1a* mutant atria. (**A–F'''**) IHC for Nkx2.5 (purple), Amhc (blue), and *fgf13a*:EGFP (SAN - green) in wild-type (WT) and *nr2f1a* mutant embryos from 48 to 96 hr post-fertilization (hpf). White arrows indicate border of Nkx2.5⁺ and *fgf13a*:EGFP⁺ cardiomyocytes. Number of embryos examined - 48 hpf: WT (n=7), *nr2f1a⁻/⁻* (n=6); 72 hpf: WT (n=7), *nr2f1a⁻/⁻* (n=10); 96 hpf: WT (n=14), *nr2f1a⁻/⁻* (n=23). Scale bar indicates 50 μm.

*Figure 4 continued on next page*

*Figure 4 continued*

The online version of this article includes the following figure supplement(s) for figure 4:

**Figure supplement 1.** Nkx2.5 is predominantly excluded from pacemaker cardiomyocytes (PCs) at the venous pole of the atrium.

**Figure supplement 2.** *Nkx2.5*:ZsYellow expression recedes toward the arterial pole of the atrium in *nr2f1a* mutants.

Cas9 system to remove the enhancer in embryos with guide RNAs (gRNAs) that flank the *nkx2.5* enhancer sequence (*Hoshijima et al., 2019*). We found that *Tg(–55nkx2.5:EGFP)* crispant (CRISPR/ Cas9-injected) embryos completely lacked or had significantly reduced EGFP expression within their hearts and other tissues where the enhancer is expressed (*Figure 6—figure supplement 2A–C*). PCR for the transgenic locus in individual *Tg(–55nkx2.5:EGFP)* embryos and the endogenous locus in individual WT embryos showed the gRNA pair used significantly abrogated both the transgenic and WT enhancer loci, respectively (*Figure 6—figure supplement 2D and E*). IHC for Nkx2.5 in *–55nkx2.5* crispants showed that Nkx2.5 did not extend as far toward to the venous pole of their atria and that they had a significant reduction in the number of Nkx2.5$^+$/Amhc$^+$ cardiomyocytes within the atria compared to control embryos (*Figure 6G–I*). Conversely, *–55nkx2.5* crispant embryos concurrently carrying the *Et(fgf13a:EGFP)* and *Tg(myl7:DsRed2-NLS)* transgenes showed an arterial-directed expansion of *Et(fgf13a:EGFP)* expression within their atria and an increase in the number of *fgf13a*:EGFP$^+$ cardiomyocytes at the venous pole compared to control embryos (*Figure 6J–L*). Altogether, these data support that this putative *nkx2.5* enhancer is required to promote or maintain Nkx2.5 expression within a subpopulation of ACs that in turn may limit the extent of PCs within the venous pole of the atrium.

## Discussion

Vertebrate embryonic cardiomyocytes maintain a high degree of plasticity, with their identity being continually reinforced even after they have overtly differentiated (*Barth et al., 2005*; *Ng et al., 2010*; *Pradhan et al., 2017*; *Tabibiazar et al., 2003*; *Targoff et al., 2013*; *van Weerd and Christoffels, 2016*; *Xin et al., 2007*). Here, our data have advanced our understanding of the requirements of Nr2f TFs within the vertebrate heart and show that they are required to maintain AC identity while concurrently repressing both VC and PC identity in different regions of the atrium (*Figure 7*). In mice, Nr2f2 maintains AC identity by repressing the expression of TFs that promote a ventricular identity program, including *Irx4* and *Hey2* (*Wu et al., 2013*). Moreover, murine Nr2f2 appears to be required over a relatively long developmental period after ACs express differentiation markers to prevent the acquisition of VC identity within the atrium. Specifically, loss of Nr2f2 by e9.5 using the *Myh6-Cre* or by e12.5 using the inducible *Myh6-MerCreMer* both result in the ectopic expression of VC differentiation genes within ACs (*Wu et al., 2013*). Conversely, ectopic Nr2f2 expression in VCs is sufficient to induce AC gene expression (*Wu et al., 2013*). Although our previous work had shown that zebrafish Nr2f1a promotes atrial differentiation (*Duong et al., 2018*), it was still unclear if ACs in zebrafish *nr2f1a* mutants acquired VC identity, similar to what was shown for *Nr2f2* loss in mice. We now show that there is acquisition of VC identity within the *nr2f1a* mutant atria. However, with the murine conditional *Nr2f2* KO approaches, the two atria were formed, but enlarged and appeared to express VC markers throughout the atria (*Wu et al., 2013*). In contrast to what was shown in the conditional KO mice, the acquisition of VC identity in zebrafish *nr2f1a* mutants was limited to the population of less-differentiated cardiomyocytes that comprise the expanded AVC, which extends into the smaller single atria. Moreover, our data show that initially within the early zebrafish heart tube there is significant overlap of Vmhc and Amhc expression. Normally, the population of cardiomyocytes expressing both Vmhc and Amhc becomes significantly refined to the AVC. However, this population of cardiomyocytes that comprise the AVC is not restricted within the *nr2f1a* mutants and they then progressively resolve to VC identity in an arterial-to-venous direction via losing expression of AC differentiation markers. We presently cannot rule out that these overt differences in effects on the ACs are due to slightly different species-specific requirements of Nr2f TFs within mice and zebrafish or technical differences from the global zebrafish *nr2f1a* mutants vs the conditional *Nr2f2* approaches. Nevertheless, taken together, the trends reinforce that there are conserved requirements for vertebrate Nr2f TFs in repressing VC identity within ACs. However, our results with global loss of Nr2f1a support that not all ACs have the potential to transdifferentiate into VCs and that this transformation occurs in a sensitized AC population.

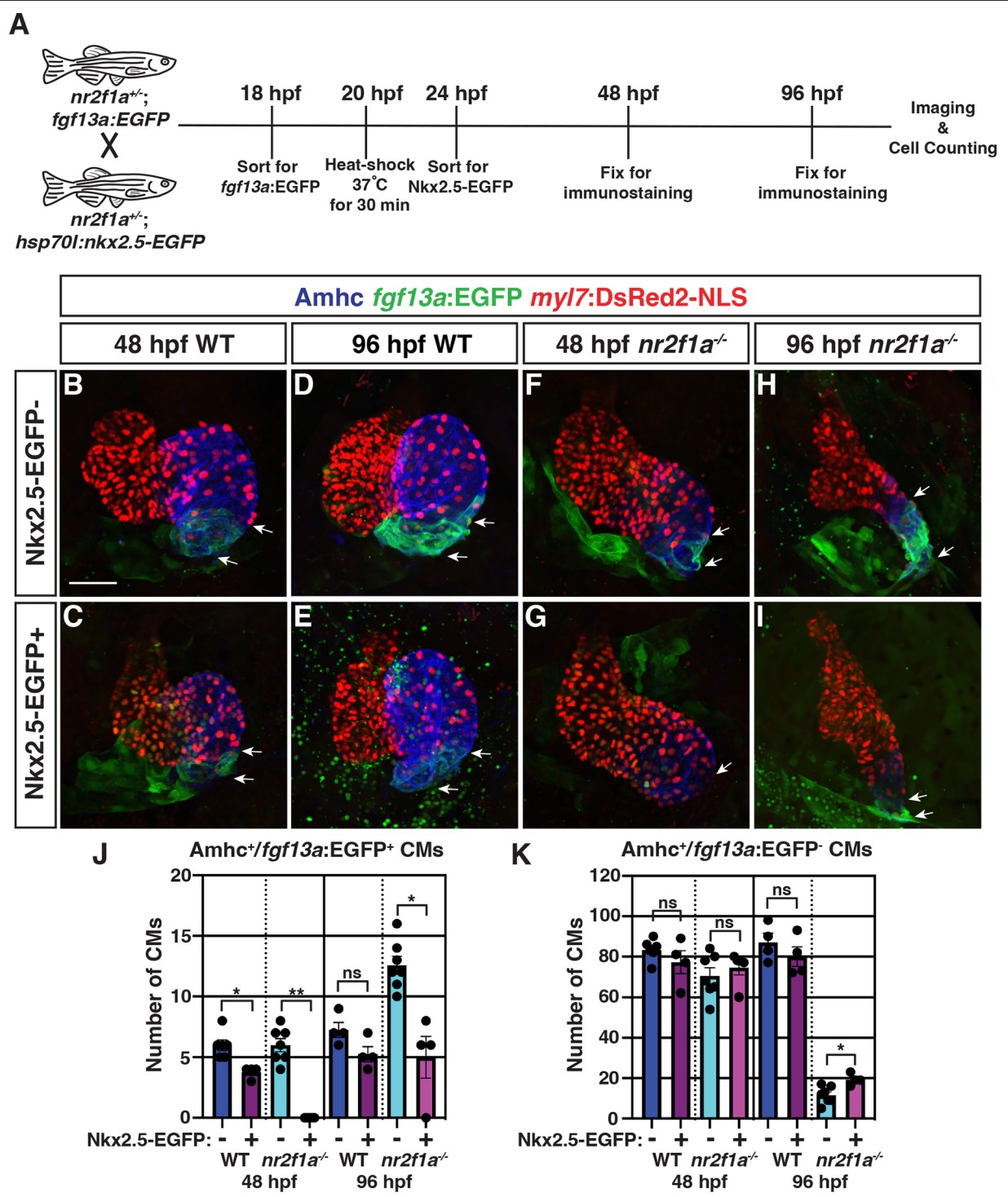

**Figure 5.** Nkx2.5 induction at 20 hr post-fertilization (hpf) represses pacemaker cardiomyocyte (PC) marker expansion in *nr2f1a* mutant atria. (**A**) Schematic of timeline for heat-shock experiments. (**B–I**) IHC of representative hearts for Amhc (blue), *fgf13a*:EGFP (green), and *myl7*:DsRed2-NLS (red) used to quantify Amhc⁺/*fgf13a*:EGFP⁺ cardiomyocytes (CMs) marking PCs from wild-type (WT) and *nr2f1a* mutant embryos with and without Nkx2.5-EGFP. White arrows indicate boundaries of *Tg(fgf13a:EGFP)* expression. Scale bar indicates 50 μm. (**J–K**) Quantification of Amhc⁺/*fgf13a*:EGFP⁺/*myl7*:DsRed2-NLS⁺ (PCs) and Amhc⁺/*fgf13a*:EGFP⁻/*myl7*:DsRed2-NLS⁺ cardiomyocytes in the atria with and without Nkx2.5-EGFP induction; 48 hpf WT: Nkx2.5-EGFP⁻ (n=6), Nkx2.5-EGFP⁺ (n=4); 48 hpf *nr2f1a⁻/⁻*: Nkx2.5-EGFP⁻ (n=7), Nkx2.5-EGFP⁺ (n=5); 96 hpf WT: Nkx2.5-EGFP⁻ (n=4), Nkx2.5-EGFP⁺ (n=4); 96 hpf *nr2f1a⁻/⁻*: Nkx2.5-EGFP⁻ (n=7), Nkx2.5-EGFP⁺ (n=4). Differences between WT and *nr2f1a⁻/⁻* at the different time points were analyzed using Student's t-test. Error bars indicate s.e.m. *p=0.05–0.001, **p<0.001.

*Figure 5 continued on next page*

*Figure 5 continued*

The online version of this article includes the following figure supplement(s) for figure 5:

**Figure supplement 1.** Induction of Nkx2.5 at 20 hr post-fertilization (hpf) does not rescue heart rate.

**Figure supplement 2.** Nkx2.5-EGFP induction at 40 hr post-fertilization (hpf) partially represses the expansion of pacemaker cardiomyocyte (PC) identity.

Our data also support a new paradigm whereby Nr2f TFs are required to limit the acquisition of PC identity within more venous ACs. Work predominantly in mice has revealed a transcriptional regulatory network that promotes the differentiation of PCs within the SAN at the venous pole of the right atrium (*Burkhard et al., 2017*; *Christoffels et al., 2010*; *van Weerd and Christoffels, 2016*). Tbx3, Isl1, and Shox/Shox2 TFs are part of a core regulatory network that promotes PC differentiation within venous ACs, while Nkx2.5 represses PC identity in working ACs (*Blaschke et al., 2007*; *Espinoza-Lewis et al., 2009*; *Espinoza-Lewis et al., 2011*; *Hoogaars et al., 2007*; *Liang et al., 2015*; *Liu et al., 2011*; *Mommersteeg et al., 2007*; *Nakashima et al., 2014*; *Sun et al., 2007*; *Weinberger et al., 2012*; *Wiese et al., 2009*). Although the characterization of PC differentiation has been less extensive in zebrafish, functional analysis of Isl1, Shox/Shox2, and Nkx2.5 support that this regulatory network is fundamentally conserved (*Blaschke et al., 2007*; *Colombo et al., 2018*; *de Pater et al., 2009*; *Hoffmann et al., 2013*; *Tessadori et al., 2012*). Our data show that in *nr2f1a* mutants there is a progressive expansion of central conduction system identity from the venous pole toward the arterial pole of the atrium, supporting a role for Nr2f TFs in regulating this conserved network within ACs. The progressive expansion, where the number of PCs is initially unchanged in *nr2f1a* mutant atria despite being smaller, contrasts some with *Nkx2.5* mutant mice and zebrafish (*Colombo et al., 2018*; *Mommersteeg et al., 2007*; *Nakashima et al., 2014*), which show an initial expansion of PC markers and ectopic differentiation at earlier stages. Although the studies of *Nr2f2* conditional KO mice did not examine PC and SAN differentiation, manual interrogation of their data, which was from whole hearts, does show they have increased expression of *Tbx3* and *Fgf13* (*Wu et al., 2013*), implying that this requirement in repressing PC identity may also be conserved within the vertebrate atrium. However, determining if murine Nr2f2 possesses similar requirements in repressing PC identity at the venous pole of the right atrium in mammals will require additional analysis.

Our data support that one mechanism by which Nr2f1a intersects with the PC regulatory network and represses the progressive acquisition of PC identity within ACs is through maintaining Nkx2.5 expression at the venous pole of the atrium. Interestingly, Nkx2.5 induction at early stages of heart development can initially completely repress PC reporter expression in the *nr2f1a* mutant atria at later stages. Thus, these results imply that the transcriptional complexes reflecting the complementary changes in Nkx2.5 and PC gene expression observed at later stages must actually be established much earlier during heart development and that Nr2f1a loss actually sensitizes these ACs to Nkx2.5. Furthermore, early Nkx2.5 induction does not produce a permanent repression of PC reporter expansion, as the PC reporter expands by 96 hpf, though less than in *nr2f1a* mutant atria. While one interpretation could be that Nkx2.5 expression needs to be maintained within the ACs over a longer period than provided by the heat-shock to prevent PC expansion, as the Nkx2.5-GFP is only thought to last ~8 hr (*George et al., 2015*), a long requirement would seem unlikely as *nkx2.5* mutants can also be rescued to adulthood following a single heat-shock at the 20 somite stage. Therefore, we postulate that Nr2f1a must also concurrently affect the AC expression of other TFs within the core PC regulatory network that promote PC differentiation, which warrants further investigation. Furthermore, it is interesting that the putative *Tg(−55nkx2.5:EGFP)* enhancer begins expression at later stages (~72 hpf), correlating with the PC marker expansion and implying the enhancer may be activated to reinforce *nkx2.5* expression at this border. Additionally, Nr2f1a appears to be expressed equivalently throughout the cardiomyocytes of the zebrafish atrium (*Duong et al., 2018*; *Figure 6— figure supplement 3*) and we did not observe overlap of VC and PC markers within the *nr2f1a* mutant atria at the stages examined, suggesting that Nr2f1a at least functionally, if not directly, must interact with other signaling pathways and factors to confer the regional effects within these AC populations. Altogether, our current data support that Nr2f1a must intersect and functionally interact with multiple gene regulatory networks to mediate its regional requirements preventing the progressive acquisition of VC and PC identity in different AC subpopulations. The nature of these factors and signaling will be investigated in the future.

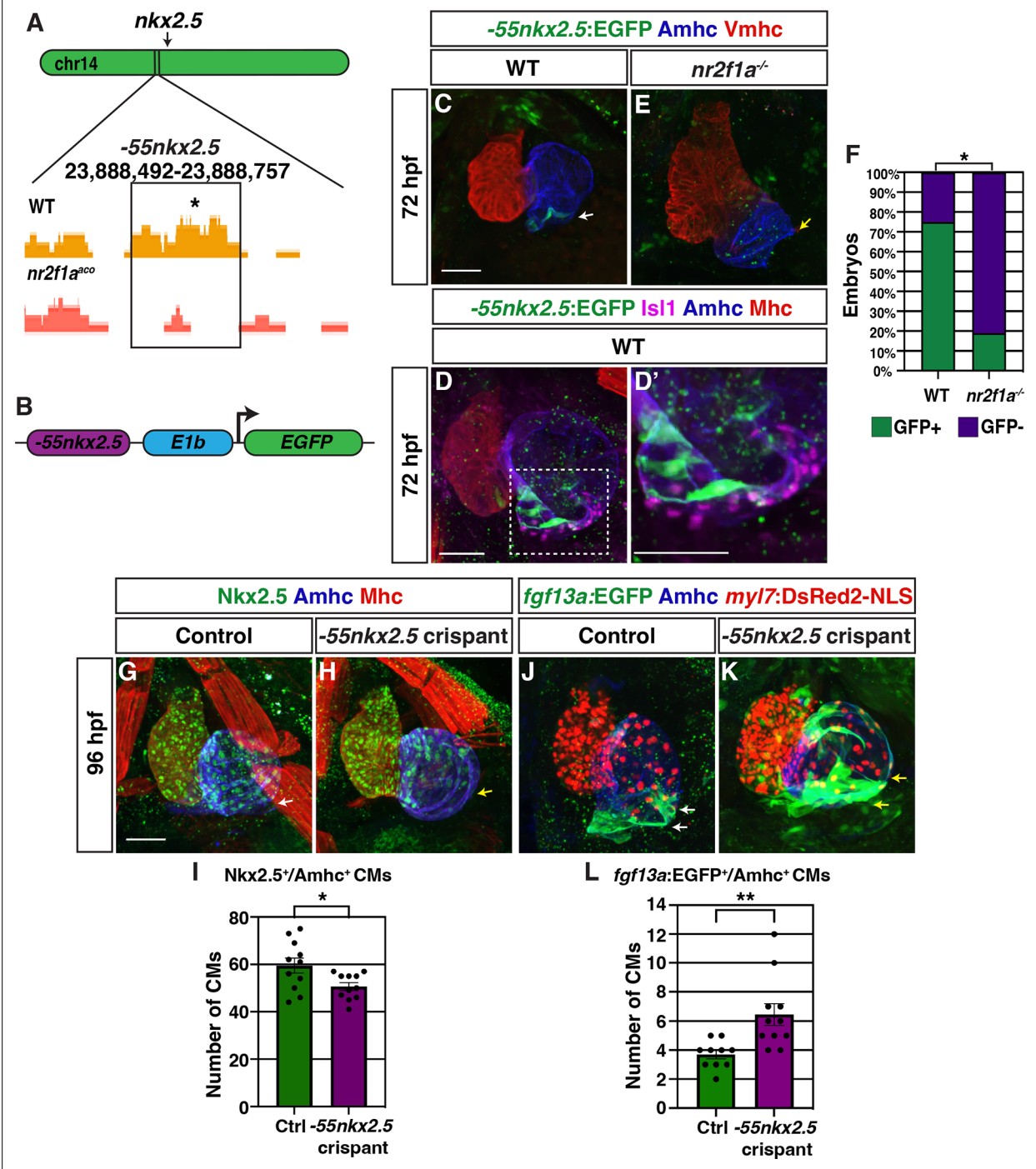

**Figure 6.** A putative *nkx2.5* enhancer is expressed in atrial cardiomyocytes (ACs) adjacent to the sinoatrial node (SAN). (**A**) Comparison of assay for transpose-accessible chromatin sequencing (ATAC-seq) peaks from wild-type (WT) and *nr2f1a* mutant *amhc*:EGFP⁺ cardiomyocytes at 48 hr post-fertilization (hpf), ~55 kb upstream of *nkx2.5*. Asterisk denotes canonical Nr2f binding site. (**B**) Schematic of –55 kb *nkx2.5* enhancer reporter construct. (**C**) IHC for *Tg(–55nkx2.5:EGFP)* enhancer (green), Amhc (blue), and Vmhc (red) in hearts at 72 hpf. White arrow indicates –55nkx2.5:EGFP expression. (**D–D'**) IHC for –55nkx2.5:EGFP (green), Isl1 (purple), Amhc (blue), and sarcomeric myosin heavy chain (Mhc: pan-cardiac - red). The transgenic enhancer reporter is expressed in the ACs directly adjacent to the Isl1⁺ pacemaker cardiomyocytes (PCs). (**E**) IHC for *Tg(–55nkx2.5:EGFP)* enhancer (green), Amhc (blue), and Vmhc (red) in an *nr2f1a* mutant heart at 72 hpf. Expression of transgenic enhancer reporter is lost in *nr2f1a* mutants. Yellow arrow indicates venous pole devoid of –55nkx2.5:EGFP expression. (**F**) Comparison of percentage of WT and *nr2f1a* mutant embryos with –55nkx2.5:EGFP expression in the heart at 72 hpf WT (n=16); *nr2f1a⁻/⁻* (n=16), using Fisher's exact test. (**G,H**) IHC for Nkx2.5 (green), Amhc (purple), and Mhc (red) in uninjected control and –55nkx2.5 crispant embryo hearts at 96 hpf. White arrow in (**G**) indicates the border of Nkx2.5⁺ cardiomyocytes extends close to the venous pole. Yellow arrow in (**H**) indicates that the Nkx2.5⁺ cardiomyocyte border is located farther from the venous pole of the atrium in crispant

*Figure 6 continued on next page*

*Figure 6 continued*

embryos. (**I**) Quantification of Nkx2.5⁺/Amhc⁺ cardiomyocytes in uninjected control (n=11) and –*55nkx2.5* crispant (n=11) embryos at 96 hpf. (**J,K**) IHC for *fgf13a*:EGFP (green), Amhc (blue), and *myl7*:DsRed2-NLS (red) in uninjected control and –*55nkx2.5* crispant *Tg(fgf13a:EGFP)⁺* embryo hearts at 96 hpf. White arrows in (**J**) indicate the region of *fgf13a*:EGFP⁺ cardiomyocytes at the venous pole. Yellow arrows in (**K**) indicates that the region of *fgf13a*:EGFP⁺ cardiomyocytes is expanded from the venous pole in the atrium of crispant embryos. (**L**) Quantification of *fgf13a*:EGFP⁺/Amhc⁺ cardiomyocytes in uninjected control (n=10) and –*55nkx2.5* crispant (n=11) embryo hearts at 96 hpf. Differences between uninjected control and –*55nkx2.5* crispants were analyzed using Student's t-test. Error bars indicate s.e.m. *p=0.05–0.001, **p<0.001. Scale bars in all images indicate 50 μm.

The online version of this article includes the following source data and figure supplement(s) for figure 6:

**Figure supplement 1.** Peaks showing open chromatin surrounding representative loci from the assay for transpose-accessible chromatin sequencing (ATAC-seq) data.

**Figure supplement 2.** Deletion of the putative *nkx2.5* enhancer in crispants.

**Figure supplement 2—source data 1.** Uncropped gel pictures of PCR analysis for efficacy of the dgRNA pairs in generating deletions of the transgenic and endogenous –*55nkx2.5* loci.

**Figure supplement 3.** Nr2f1a is expressed throughout cardiomyocytes in the atrium.

In considering the implications of our data, it is interesting that in humans, congenital malformations, including AVSDs, are associated with increased incidence of arrhythmias (*Williams and Perry, 2018*). Moreover, the arrhythmias can arise from complications secondary to the structural defects or pleiotropic requirements in the development of the myocardium and the conduction system (*Bruneau et al., 1999*; *Ellesøe et al., 2016*; *Williams and Perry, 2018*). A prime example of the latter is mutations in *NKX2.5*, which are associated with numerous CHDs, including atrial and ventricular septal defects, and conduction defects, consistent with its requirements in working cardiomyocyte and PC differentiation shown in experimental models (*Benson et al., 1999*; *Ellesøe et al., 2016*; *Elliott et al., 2003*; *Jhaveri et al., 2018*; *McElhinney et al., 2003*; *Schott et al., 1998*; *Xie et al., 2013*; *Yu et al., 2014*). Similar to *NKX2.5*, *NR2F2* mutations in humans have now been associated with myriad CHDs, including atrial septal defects and AVSDs (*Al Turki et al., 2014*; *Nakamura et al., 2011*; *Poot et al., 2007*; *Qiao et al., 2018*; *Upadia et al., 2018*). While it is presently unclear if there are also accompanying arrhythmias or conduction defects in these patients, given our results, we speculate that further evaluation of patients with *NR2F2* mutations will show that they have or are at higher risk to develop arrhythmias.

In conclusion, mutations in genes, such as *NR2F2*, that are required for the maintenance of cardiomyocyte identity have been associated with multiple types of CHDs in humans (*Al Turki et al., 2014*; *Benjamin et al., 2018*; *Benson et al., 1999*; *Cheng et al., 2011*; *Hoffman and Kaplan, 2002*; *Loffredo,*

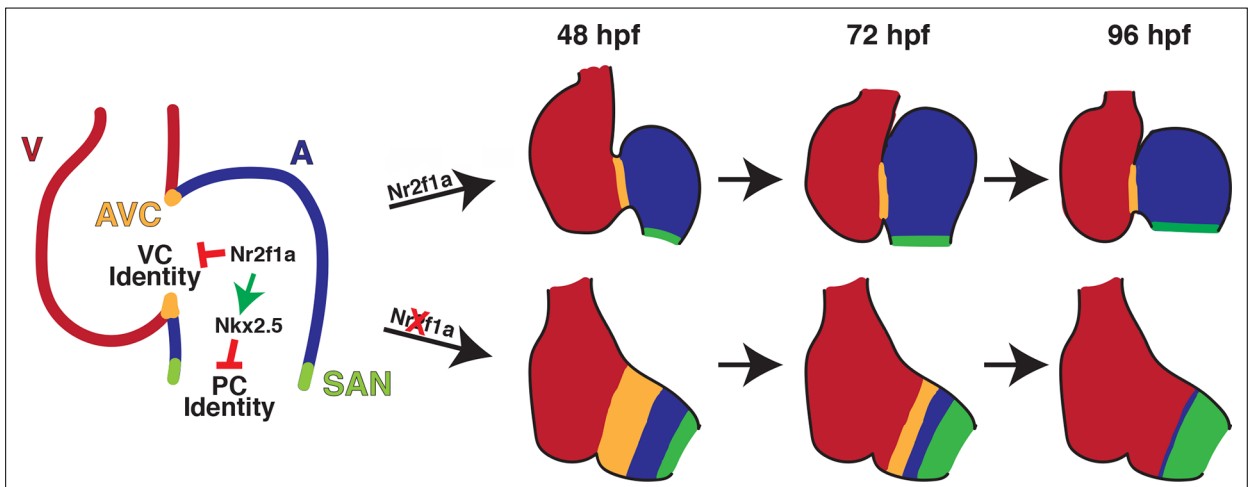

**Figure 7.** Model summarizing the consequences of Nr2f1a loss on atrial development. Following its initial requirement promoting atrial cardiomyocyte (AC) differentiation (*Duong et al., 2018*), Nr2f1a is required to maintain AC identity within the arterial (outflow) and venous (inflow) regions of the zebrafish atrium. In the absence of Nr2f1a, cardiomyocytes within the enlarged atrioventricular canal (AVC), where there is expression of both AC and ventricular cardiomyocyte (VC) differentiation markers, progressively lose expression of AC differentiation markers and only express VC differentiation markers. Concurrently, more venous ACs progressively gain PC differentiation marker expression.

*2000*; *Nakamura et al., 2011*; *Schott et al., 1998*; *van der Linde et al., 2011*). Hence, elucidating the transcriptional mechanisms that maintain cardiomyocyte identity and regulate cardiomyocyte plasticity may provide insights into the etiology of CHDs, as well as help refine stem cell-derived tissue engineering and regenerative strategies. Our results have delineated novel requirements for Nr2f TFs in maintaining AC identity in vertebrates, which may help provide a framework for our understanding of the molecular etiology of concurrent structural malformations and conduction defects in patients.

# Materials and methods

## Zebrafish lines used

Adult zebrafish were raised and maintained under standard laboratory conditions (*Westerfield, 2000*). Transgenic lines used: *Tg(–5.1myl7:DsRed2-NLS)$^{f2}$* (*Mably et al., 2003*), *SqET33-mi59B* (*Poon et al., 2016*), *TgBAC(–36nkx2.5:ZsYellow)$^{fb7}$* (*Zhou et al., 2011*), *Tg(myh6:EGFP)$^{s958}$* (*Zhang et al., 2013*), *Tg(hsp70l:nkx2.5-EGFP)$^{fcu1}$* (*George et al., 2015*), *Tg(myl7:EGFP)$^{f1}$* (*Huang et al., 2003*), *Tg(myl7:NLS-KikGR)* (*Lazic and Scott, 2011*). Three *nr2f1a* mutant alleles were used: *nr2f1a$^{el512}$* and *nr2f1a$^{ci1009}$*, which were reported previously (*Duong et al., 2018*), and *nr2f1a$^{ci1017}$*, which was identified from an ENU screen. Additional details of this allele will be reported elsewhere (manuscript in preparation). The *nr2f1a$^{ci1017}$* allele was used for RNA-seq and ATAC-seq. The *nr2f1a$^{el512}$* and *nr2f1a$^{ci1009}$* alleles were used for the photoconversion assay. All other experiments were completed using the *nr2f1a$^{ci1009}$* allele. WT and *nr2f1a$^{+/-}$* embryos were overtly indistinguishable, including with respect to their hearts. Therefore, unless otherwise indicated, WT in figures in which embryos were genotyped refers to either WT or *nr2f1a$^{+/-}$* embryos. For all experiments, embryos were grown in 0.003% 1-phenyl-2-thiourea to prevent pigmentation.

## Generation of transgenic *Tg(-55nkx2.5:EGFP)* enhancer line

The putative *nkx2.5* enhancer transgenic line was generated using standard Tol2/Gateway methods (*Kwan et al., 2007*). The enhancer region at ~55 kb upstream of *nkx2.5* was amplified with PCR of genomic DNA and cloned into the pDONR221 middle entry vector. Subsequently, it was transferred into the *E1b-GFP-Tol2* Gateway destination vector (Addgene plasmid #37846, *Birnbaum et al., 2012*). To generate transgenic lines, one cell embryos were co-injected with 25 pg of the *–55nkx2.5:EGFP* plasmid and 25 pg Tol2 mRNA (*Kwan et al., 2007*). Injected embryos were raised to adulthood and transgenic founders were identified via outcrossing to WT fish. Multiple founders with overtly equivalent GFP expression within the atrium were identified. The line that was used in this study has been designated *Tg(–55nkx2.5:EGFP)$^{ci1015}$*.

## Generation of *–55nkx2.5* enhancer crispants

To generate *–55nkx2.5* enhancer crispants (CRISPR/Cas9-injected embryos), crRNAs flanking the endogenous enhancer sequence were designed using IDT's custom gRNA design tool (https://www.idtdna.com/). crRNA:tracrRNA duplexes were prepared as described previously (*Hoshijima et al., 2019*). Briefly, equal volumes of 100 µM target-specific crRNA and common tracrRNA were mixed together and annealed by heating to 95°C for 5 min, then cooled at 0.1 °C/s to 25°C, followed by rapid cooling to 4°C in a PCR machine to make a 50 µM solution of the crRNA:tracrRNA duplex, referred to as the duplex guide RNA (dgRNA). An equal volume of duplex buffer (IDT) was then added to further dilute the dgRNA to 25 µM. To generate 5 µM dgRNA;Cas9 complexes, we combined: 1 µL of each 25 µM dgRNA, 0.4 µL 61 µM Cas9 stock (Alt-R S.p. Cas9 Nuclease V3, IDT), 1.6 µL H$_2$O, and 1 µL phenol red. Prior to injection, the solution was incubated at 37°C for 5 min and then placed at room temperature. We injected 1 nL of the 5 µM dgRNA:Cas9 complex into one-cell stage embryos. Efficiency of dgRNA:Cas9 complexes was assessed using PCR on individual embryos.

## IHC and cell quantification

IHC was performed as previously described (*Waxman et al., 2008*). Embryos were fixed in 1% formaldehyde in PBS for 1 hr at room temperature and then washed 1× in PBS, 2× in 0.1% saponin/1× PBS followed by blocking in saponin blocking solution (0.1% saponin, 10% sheep serum, 1× PBS, 2 mg/mL BSA) for 1 hr. Primary antibodies were diluted in saponin blocking solution and applied to embryos overnight at 4°C. Embryos were washed with 0.1% saponin/1× PBS multiple times before

being incubated with secondary antibodies diluted in saponin blocking solution for 2 hr at room temperature. Embryos were washed multiple times with 0.1% saponin/1× PBS before imaging. Antibody information is provided in *Supplementary file 2*.

IHC for quantification of cardiomyocytes in the AVC required the use of two rabbit primary antibodies (anti-DsRed and anti-Vmhc; *Song et al., 2019*). To accommodate this, after blocking, embryos were first incubated with anti-Amhc (anti-Myh6/S46, mouse IgG1) and anti-DsRed (rabbit) overnight at 4°C. Secondary antibodies (anti-mouse IgG1-Dylight 405 and anti-rabbit-TRITC) were then applied for 2 hr at room temperature. After washing multiple times with 0.1% saponin/1× PBS, embryos were again incubated in saponin blocking solution for 1 hr. Anti-Vmhc (rabbit) was then diluted in saponin blocking solution and applied to embryos overnight at 4°C. Embryos were then washed multiple times with 0.1% saponin/1× PBS. Anti-rabbit-Alexa-488 was diluted in saponin blocking solution and embryos were incubated in secondary antibody for 1 hr at room temperature, followed by multiple washes with 0.1% saponin/1× PBS.

IHC for activated Caspase 3 was performed as previously described (*Rydeen and Waxman, 2016*). Embryos were fixed in 4% paraformaldehyde overnight at 4°C then dehydrated in a methanol series and left in 100% methanol overnight at –20°C. Embryos were then rehydrated in PBST (PBS +0.1% Tween-20), permeabilized for 20 min using PDT (PBST + 1% DMSO) supplemented with 0.3% Triton-X, and then incubated in blocking solution (1× PBS, 10% sheep serum, 0.1% Tween-20) for 1 hr. Primary antibodies were diluted in blocking solution and applied to embryos overnight at 4°C. Embryos were then washed 3× for 20 min with PDT and blocked again in blocking solution for 30 min. Secondary antibodies were diluted in blocking solution and applied to embryos for 2 hr at room temperature. Following secondary antibody incubation, embryos were washed multiple times with PDT.

Following IHC, embryos were imaged on a Nikon A1R inverted confocal microscope. Nikon's Denoise-AI was employed on images taken using an HD resonance scanner. Cardiomyocytes were counted using Photoshop and ImageJ. For quantification of PCs, DsRed2-NLS$^+$/Amhc$^+$/*fgf13a*:EGFP$^+$ cardiomyocytes were counted as PCs, while the DsRed2-NLS$^+$/Amhc$^+$/*fgf13a*:EGFP$^-$ cardiomyocytes were counted as atrium. For AVC counting, DsRed2-NLS$^+$/Amhc$^+$/Vmhc$^-$ cardiomyocytes were counted as atrium, DsRed2-NLS$^+$/Amhc$^+$/Vmhc$^+$ cardiomyocytes were counted as AVC, and DsRed2-NLS$^+$/Amhc$^-$/Vmhc$^+$ cardiomyocytes were counted as ventricle.

## EdU assay for cell proliferation

EdU labeling was performed using a Click-iT EdU Alexa Fluor Imaging Kit (Molecular Probes) according to the manufacturer's instructions. At 48 or 72 hpf, embryos carrying the *Tg(myl7:DsRed2-NLS)* transgene were placed in glass vials (30 embryos per vial) and incubated with 10 mM EdU for 30 min on ice. EdU was then washed out and embryos were returned to Petri dishes and incubated at 28.5°C until 72 or 96 hpf, respectively. Embryos were then fixed in 1% formaldehyde in PBS and IHC was performed using saponin protocol listed above. After IHC, embryos were post-fixed in 2% formaldehyde in PBS for 1 hr at room temperature followed by washes with 1× PBS. The Click-iT reaction was then performed according to the manufacturer's protocol. Embryos were imaged on a Nikon A1R inverted confocal, as described above. In each chamber, the number of EdU$^+$/DsRed2-NLS$^+$ cardiomyocytes were counted and normalized to the total number of DsRed2-NLS$^+$ cardiomyocytes to determine the proliferation index.

## Cardiomyocyte temporal differentiation assay

Embryos from a cross of adult *nr2f1a$^{+/-}$; Tg(myl7:NLS-KikGR)* zebrafish and *nr2f1a$^{+/-}$; Et(fgf13a:EGFP)* zebrafish were sorted for both the *Et(fgf13a:EGFP)* and *Tg(myl7:NLS-KikGR)* transgenes. As the *Et(fgf13a:EGFP)* is expressed in the PCs and the *Tg(myl7:NLS-KikGR)* is expressed in all cardiomyocytes, we were able to distinguish between these two transgenes before photoconversion of the NLS-KikGR. At 72 hpf, embryos carrying both transgenes were anesthetized in 0.16 mg/mL tricaine and exposed to UV light using a DAPI filter on Zeiss M2BioV12 Stereo microscope for 30 min to photoconvert the NLS-KikGR. At 96 hpf, photoconverted embryos were fixed in 1% formaldehyde in PBS for 1 hr at room temperature, then washed 1× in PBS followed by 2× in 0.1% saponin/1× PBS. Embryos were imaged the same day they were fixed using a Nikon A1R inverted confocal microscope with an HD resonance scanner. Nikon's Denoise-AI was employed on acquired images.

## Heart rate analysis

To analyze heart rate, embryos were anesthetized in 0.16 mg/mL tricaine and mounted in 1% low-melt agarose. Hearts were imaged on a Nikon Ti-2 SpectraX Widefield microscope with an Andor Xyla 4.2 megapixel, 16-bit sCMOS monochromatic camera at 200 frames per second (fps) in a controlled temperature chamber (28.5°C). Analysis performed in Nikon Elements software.

## Heat-shock experiments

Embryos resulting from a cross of adult *nr2f1a*^+/-^; *Et(fgf13a:EGFP)*; *Tg(myl7:DsRed2-NLS)*; and *nr2f1a*^+/^; *Tg(hsp70l:nkx2.5-EGFP)* zebrafish were first sorted for the *Et(fgf13a:EGFP)* transgene, as it is also expressed at low levels in the epidermis and in the olfactory pits (*Poon et al., 2016*). At 20 or 40 hpf, 10 *Et(fgf13a:EGFP)*^+^ embryos per tube were placed into 0.5 mL PCR tubes with 200 µL embryo water and heat-shocked for 30 min by raising the temperature to 37°C using a thermocycler. Embryos were then returned to Petri dishes and placed in a 28.5°C incubator. 3 to 4 hr after the heat-shock, when the Nkx2.5-EGFP becomes visible, embryos were sorted based on the presence of Nkx2.5-EGFP, which could be distinguished over the fluorescence from the *Et(fgf13a:EGFP)* transgene as the Nkx2.5-EGFP signal is much stronger than that of the *Et(fgf13a:EGFP)*. Nkx2.5-EGFP is not visible by 10 hr post heat-shock (*George et al., 2015*) and, as such, does not interfere with quantification of *fgf13a:*EGFP^+^ cardiomyocytes. Embryos were then returned to the 28.5°C incubator and allowed to develop until the desired timepoints when they were harvested, processed for IHC, imaged with confocal microscopy, and the cardiomyocytes counted as described above. Embryos without the *Tg(hsp70l:nkx2.5-EGFP)* transgene that were heat-shocked served as controls. All embryos were genotyped for the *nr2f1a* mutant allele as well as the *Tg(hsp70l:nkx2.5-EGFP)* transgene following analysis, to ensure embryos were sorted correctly after the heat-shock.

## Fluorescence-activated cell sorting

EGFP^+^ cardiomyocytes from WT and *nr2f1a* mutant with the *Tg(amhc:EGFP)*, *Tg(myl7:EGFP)*, or *Tg(amhc:EGFP)*; *Tg(myl7:DsRed2-NLS)* transgenes were isolated using FACS at 48 or 96 hpf, as indicated. Briefly, embryos were dissociated as described previously (*Holowiecki et al., 2020*; *Samsa et al., 2016*; *Stachura and Traver, 2011*) with the following modifications. WT and *nr2f1a* mutant embryos were harvested, rinsed in Hank's Balanced Salt Solution with Ca^2+^ and Mg^2+^ and transferred to Eppendorf tubes in 500 µL of FACS media (0.9× PBS, 2% fetal bovine serum). 60 µL of Liberase (Roche) was added and a pipette was used to triturate the embryos. Embryos were then incubated at 32.5°C for 15 min before being triturated again. This process was repeated until the embryos were completely dissociated. Following dissociation, the cell suspensions were centrifuged at 1300 rpm (100 × *g*) for 5 min at 4°C. The supernatant was discarded, and the pellets were resuspended in a total volume of 400 µL. Samples were then filtered through a 40 µm strainer and collected into a FACS tube. FACS was performed by the CCHMC Research Flow Cytometry Core using a 70 µm nozzle and cells were collected into Eppendorf tubes for downstream applications.

## Real-time qPCR

2229 WT and 704 *nr2f1a* mutant *Tg(myl7:EGFP)*^+^ were captured using flow-sorting at 96 hpf. PolyA RNA was collected from the cardiomyocytes using the Single Cell RNA Purification Kit (Norgen Biotek) and then amplified using the SeqPlex RNA Amplification Kit (Sigma-Aldrich), which generates double-stranded cDNA. RT-qPCR was performed using SYBR Green Mix in a Bio-Rad CFX-96 PCR machine. Expression levels were standardized to β-actin and analyzed using the $2^{-\Delta\Delta CT}$ Livak method. Primer sequences are provided in *Supplementary file 3*.

## RNA-seq and ATAC-seq analysis

For RNA-seq, 7797 WT and 4118 *nr2f1a* mutant *Tg(amhc:EGFP)*^+^ were captured from flow-sorting of ~200 WT and *nr2f1a* mutant embryos at 48 hpf. RNA was collected using the Single Cell RNA Purification Kit (Norgen Biotek). PolyA RNA was then amplified using the MessageAmpII aRNA Amplification Kit (Thermo Fisher) and submitted to the CCHMC Sequencing core for quality control and library generation. Single-end sequencing was performed. RNA-seq data were submitted to the CCHMC Bioinformatics (BMI) core for analysis. To assess the need for trimming of adapter sequences and bad quality segments, the RNA-seq reads in FASTQ format were first subjected to quality control using

FastQC v0.11.5, Trim Galore! V0.4.2, and Cutadapt v1.9.1 (*Andrews, 2010*; *Krueger, 2012*; *Martin, 2011*). The trimmed reads were aligned to the reference zebrafish genome version GRCz10/danRer10 with the program STAR v2.5.2 (*Dobin et al., 2013*). Aligned reads were stripped of duplicate reads using Picard v1.89 (*Broad Institute, 2018*). Gene-level expression was assessed by counting features for each gene, as defined in the NCBI's RefSeq database (*Li et al., 2009*). Read counting was done with the program feature Counts v1.5.3 from the Rsubread package (*Liao et al., 2019*). Raw counts were normalized as transcripts per million. Differential gene expressions between conditions were assessed with the R v3.4.4 package DESeq2 v1.18.1 (*Love et al., 2014*).

For ATAC-seq, 8230 WT and 4338 *nr2f1a* mutant *Tg(amhc:EGFP)*[+]; *Tg(myl7:DsRed2-NLS)*[+] were captured from flow-sorting of ~200 WT and *nr2f1a* mutant embryos at 48 hpf. ATAC-seq libraries were generated from the cells using the Nextera Kit (Illumina), as previously reported (*Buenrostro et al., 2015*). Libraries were submitted to the CCHMC Sequencing core for paired-end sequencing. ATAC-seq data were then submitted to the CCHMC BMI Core for analysis. Reads in FASTQ format were subjected to quality control, as above. The trimmed reads were aligned to the zebrafish genome (GRCz10/danRer10) using STAR v2.5.2 with splice awareness turned off. Aligned reads were stripped of duplicate reads using Picard v1.89. Peaks were called using MACS v2.1.2 using the broad peaks mode (*Zhang et al., 2008*). Peaks with q-value <0.01 were selected for further analysis. Common peaks among all samples were obtained by merging called peaks at 50% overlap using BEDtools v2.27.0 (*Quinlan and Hall, 2010*). The common peaks, originally in BED format, were converted to a Gene Transfer Format to enable fast counting of reads under the peaks with the program feature Counts v1.5.3 (Rsubread package) (*Liao et al., 2019*). Differential open chromatin relative to control samples was assessed with the R package DESeq2 v1.18.1 (*Love et al., 2014*). Each differentially open chromatin region was annotated with closest differentially expressed gene from the RNA-seq. Differentially open chromatin regions are selected for motif enrichment analysis using HOMER v4.10 (*Heinz et al., 2010*) and known zebrafish motifs from CIS-BP v1.92 (*Weirauch et al., 2014*) are used to detect motif occurrences.

RNA-seq and ATAC-seq data used are deposited in GEO - accession number GSE194054.

## Electrophysiological analysis of hearts

Optical mapping of action potential duration and conduction velocities were performed similar to what was previously reported (*Colombo et al., 2018*; *Mosimann et al., 2015*; *Panáková et al., 2010*). Briefly, hearts from zebrafish embryos were isolated at 48, 72, and 96 hpf and stained with the transmembrane potential-sensitive dye FluoVolt (Invitrogen) for 20 min. To measure action potential duration, fluorescence intensities were recorded with a high-speed CCD camera (Redshirt Imaging) with 14-bit resolution at 2000 fps. Action potentials were temporally (800 Hz low-pass cutoff) and spatially (3×3 pixel average) filtered to enhance signal-to-noise ratios. Action potential duration at 20% repolarization (APD20) was defined as the absolute time difference between the decay phase and the upstroke phase of the local action potential at 20% repolarization and was measured at the midpoint of the atria. Conduction velocity vectors across the hearts were derived using an established algorithm (*Bayly et al., 1998*) with custom scripts in MATLAB (Version R2018B, Mathworks) and estimated from the local action potentials (*Panáková et al., 2010*). All electrophysiological measurements were averaged across the relevant anatomical segments for each heart. Embryos were genotyped following analysis.

## Statistical analysis

Comparisons between groups were analyzed using Student's t-test and one- and two-way ANOVAs with multiple comparisons as appropriate. Fisher's exact test was used to determine if two proportions were statistically distinct. Statistical analysis used for the different analyses is indicated in the figure legends. Statistical analysis was performed using GraphPad Prism. A p-value of <0.05 was considered statistically significant for all analysis. Error bars indicate s.e.m.

## Materials availability

Materials created that were used in this study are available from the authors without restrictions.

## Acknowledgements

We thank members of the Waxman lab for fruitful discussions and the CCHMC veterinary staff for care of the zebrafish. This work was supported by funding from the National Institutes of Health: [R01 HL137766, R01 HL141186, and R01 HL154522 to JSW], [F31 HL152600 to KEM], and [R24 OD017870 and the Leducq Foundation to CAM].

## Additional information

### Funding

| Funder | Grant reference number | Author |
| --- | --- | --- |
| National Heart, Lung, and Blood Institute | R01 HL137766 | Joshua S Waxman |
| National Heart, Lung, and Blood Institute | R01 HL141186 | Joshua S Waxman |
| National Heart, Lung, and Blood Institute | R01 HL154522 | Joshua S Waxman |
| National Heart, Lung, and Blood Institute | F31 HL152600 | Kendall E Martin |
| NIH Office of the Director | R24 OD017870 | Calum A MacRae |
| Leducq Foundation | | Calum A MacRae |

The funders had no role in study design, data collection and interpretation, or the decision to submit the work for publication.

### Author contributions

Kendall E Martin, Conceptualization, Data curation, Formal analysis, Funding acquisition, Validation, Investigation, Visualization, Methodology, Writing – original draft, Writing – review and editing; Padmapriyadarshini Ravisankar, Formal analysis, Investigation, Writing – review and editing; Manu Beerens, Data curation, Formal analysis, Validation, Investigation, Visualization, Methodology, Writing – review and editing; Calum A MacRae, Resources, Supervision, Funding acquisition, Investigation, Visualization, Writing – original draft, Writing – review and editing; Joshua S Waxman, Conceptualization, Resources, Supervision, Funding acquisition, Investigation, Visualization, Writing – original draft, Project administration, Writing – review and editing

### Author ORCIDs

Joshua S Waxman ⓘ http://orcid.org/0000-0002-8132-487X

### Ethics

All zebrafish husbandry and experiments were performed in accordance with protocols approved by the Cincinnati Children's Hospital Medical Center Institutional Animal Care and Use Committee (IACUC). Protocol # 2020-0091.

### Decision letter and Author response

Decision letter https://doi.org/10.7554/eLife.77408.sa1
Author response https://doi.org/10.7554/eLife.77408.sa2

## Additional files

### Supplementary files

• Supplementary file 1. (xlsx) Sheets with integrated and primary RNA-seq and assay for transpose-

accessible chromatin sequencing (ATAC-seq) data with differential gene expression, changes in called peaks, and putative Nr2f binding sites.

- Supplementary file 2. Antibody information.
- Supplementary file 3. Primer information.
- MDAR checklist

### Data availability
Sequencing data have been deposited in GEO - accession number GSE194054.

The following dataset was generated:

| Author(s) | Year | Dataset title | Dataset URL | Database and Identifier |
|---|---|---|---|---|
| Martin KE, Ravisankar P, Beerens MM, MacRae CA, Waxman JS | 2022 | Nr2f1a maintains atrial nkx2.5 expression to repress pacemaker identity within venous atrial cardiomyocytes of zebrafish | https://www.ncbi. nlm.nih.gov/geo/ query/acc.cgi?acc= GSE194054 | NCBI Gene Expression Omnibus, GSE194054 |

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
