## [Editor Report]

Addressing the role of NR2F transcription factors in the fish heart, this important work provides novel insight into atrial chamber patterning. Solid evidence supports the identification of a new mechanism of pacemaker cell restriction, involving nkx2.5 maintenance in the atrium by nr2f1a. Whether nr2f1 also restricts ventricular fate as shown in the mouse model will require more investigation. This manuscript will be of interest to developmental biologists and pediatric cardiologists.

---

## [Decision Letter]

**Decision letter after peer review:**

Thank you for submitting your article "Nr2f1a maintains atrial nkx2.5 expression to repress pacemaker identity within venous atrial cardiomyocytes" for consideration by *eLife*. Your article has been reviewed by 3 peer reviewers, including Sigolène M Meilhac as the Reviewing Editor and Reviewer #1, and the evaluation has been overseen by Edward Morrisey as the Senior Editor. The following individuals involved in the review of your submission have agreed to reveal their identity: Vincent M. Christoffels (Reviewer #3).

Essential revisions:

As you will see, reviewers refer to a strong paper addressing patterning of the atrial chamber and a striking new mechanism of pacemaker cell restriction, but they also indicate that further work would be necessary for publication. In particular, three major issues have been raised.

1) The claim of transdifferentiation is currently not supported by the data. The definition of AC/VC/PC requires clarification, as well as that of the boundary between chambers. The same definition should be used throughout the paper. The tracking of a limited number of markers (fgf13a:GFP transgene for example) to ascribe cell lineage or inconsistencies between markers (cells that take on ventricular gene expression in mutants have more of an atrial electrophysiological character) raise concerns about the conclusion of transdifferentiation. Inconsistencies between figures (1D versus 1J-J, 2F versus 4F, 1B and 2K, 1J-L versus S1 and 2G-I) should be resolved. An in-depth analysis of the ATAC-seq and RNA-seq data in wild-type and mutant atrial cells and comparison with epigenetic state data from other models would provide better insights into lineage changes. In addition, cell tracking is required to conclude transdifferentiation.

2) Alternative mechanisms through which pacemaker cells are expanded should be addressed experimentally: cell death, cell proliferation, or progenitor recruitment.

3) The identification of nkx2.5 as a target of nr2f1a to repress pacemaker cells needs strengthening. Please explain how nrf2f1a is envisioned to act as a suppressor of pacemaker phenotype (atrial-specific activator of nkx2-5) as the gene is expressed in pacemaker cells and in the posterior second heart field progenitors the atria and sinus venosus (including pacemaker cells) derive from. Maintenance of nkx2.5 expression by nr2f1a at the venous pole is incongruent with the atrial to ventricular shifts observed. More details on nr2f1a expression and co-expression with nkx2.5 and pacemaker markers would be useful. The authors assume the identified enhancer regulates nkx2-5, but do not provide evidence for this. Overexpression of nkx2.5 not completely suppresses but delays the appearance of pacemaker cells. Examination of nkx2.5 rescue should be extended at a functional level (cardiac function and electrophysiology).

Detailed comments of the reviewers are below. Please take into account all comments and provide more explanations on the methods as detailed by the reviewers. If you are not able to address experimentally some reviewer comments within the revision period, please discuss the limitations or alternative hypotheses in the manuscript.

*Reviewer #1 (Recommendations for the authors):*

– Please provide a clear indication of the biological system under investigation in the title.

– Abbreviations complicate reading.

– How fluorescence of Nkx2.5-EGFP is distinguished over the Et(fgf13a:EGFP) transgene?

– Was cell counting done in 3D or in 2D in a projection image? Can a partial cell count in 2D explain variations between Fig2G+H and 1J? Please clarify the method section.

– Please complete the criteria of differential expression in the RNA-seq (p-value or adjusted p-value ? cut-off ?).

– "ectopic expression" (line 139) is not correct since amhc positive cells normally do express ventricular and pacemaker markers.

– Two mutant lines are described in the method section. Which line is used for each experiment? How have they been validated for the loss of nr2f1a expression?

– How are vmhc and AVC markers in the transcriptomic dataset?

– Provide single-channel images for Figure 1C-H, 2A-F to clarify marker overlap.

*Reviewer #2 (Recommendations for the authors):*

1. Figure 1: the overlap in Amhc and Vmhc should be more clearly shown. As there appears to be a significant overlap by 48 hpf (1D), what do these markers show at earlier time points? Finally, the image shown in 1D does not appear to be congruent with the data in 1J-L – based on the image, I would expect to see more vmhc + CMs in the mutant atrium at 48 hpf.

2. Line 168: The cell death and proliferation data are important to argue it is a fate conversion vs loss/expansion of populations that is occurring. It should be shown, or some form of fate tracing (photoconversion, etc.) should be used if statements like those in lines 174/5 are to be made.

3. Figure 4: The extent of fgf13a:GFP + cells in the atrium in nr2f1a mutants at 96hpf appears to differ markedly in Figures 2 (most of the cells are +) vs 4 (appears to be a lower proportion of the atrium as shown). How are chamber boundaries being defined in these analyses?

4. Figure 5: Somewhat confusing – how are new PCs made from 48 to 96hpf if Nkx2.5 is still present? ACs are still lost in "rescued" hearts – presumably, these now make VCs? Importantly, does the nkx2.5 transgene rescue/normalize heart rate or conduction? It is important to distinguish an effect on the fgf13a transgene (which is convincing) versus a more important effect on PC and AC fate and FUNCTION is reflected in these results.

*Reviewer #3 (Recommendations for the authors):*

1) Please provide more details in figure 1 regarding the RNAseq analysis. Why did the authors use a cut-off of >2 fold difference rather than significance? Could they provide an MA plot or so to show distributions of FC and p values of all transcripts detected? How many transcripts were differentially expressed? What kind of biological processes were changed?

2) In figure 1B the behavior of pacemaker markers is inconsistent. Please provide data or references showing the expression of these markers in fish pacemaker cells (e.g. I did not find these data for tbx3a etc.). Furthermore, central pacemaker markers like isl1a (Isl1 homologue) and shox2a (Shox2 homologue) are downregulated in mutants? That is unexpected, and also inconsistent with the data given in figure 2K.

3) Arterial pole of the atrium is confusing, as arterial pole usually refers to the actual arterial pole of the heart (distal/downstream portion of the ventricle).

4) Please explain how the authors envision nrf2f1a/Nrf2f2 acts as a suppressor of pacemaker phenotype (atrial-specific activator of nkx2-5) as the gene is expressed in pacemaker cells and in the posterior second heart field progenitors the atria and sinus venosus (including pacemaker cells) derive from.

5) The cell numbers seem inconsistent between the different figures. In suppl Figure 1, there are about 75 amhc+ cells in wt, and 20 or so in mutants. In figure 1, there are close to 100 amhc+ cells in wt, and about 50 in mutants? In figure 2 there are 60 amhc+ cells in wt, and 40, 20 and 10 amhc+ cells in mutants at subsequent stages. If one adds the amhc+ and fgf13a-gfp+ cells, the numbers are 40, 30, 20, respectively (figure 2c). Which numbers are correct? Is the number of amhc+ cells very low in mutants from stage 48 hpf onwards, without changing, or is there a decrease during development?

A more general issue related to the cell number issues, the data can be explained by transdifferentiation of atrial cells into pacemaker cells, as the authors seem to favor, or by reduced addition of cardiomyocytes fated to become atrial, while the normal atrial-pacemaker demarcation programs are still active, or by reduced atrial differentiation. The latter 2 options are consistent with the data of suppl Figure 1. The authors should resolve this, and provide additional support for the trans-differentiation hypothesis. Thus, does a fraction of cells expressing pacemaker markers derive from atrial marker expressing cells in mutants, or is there an increased population of "de novo" pacemaker cells (not previously atrial marker positive) in the venous pole and a decreased population of atrial cells because of defective recruitment from progenitors or defective atrial differentiation.

6) Line 201: "Thus, we would predict…" Why would the heart rate be predicted to decrease? More pacemaker cells in mutants may also result in increased automaticity and increased spontaneous depolarizations.

7) The relation between hyperpolarization, increased repolarization times and increased numbers of pacemaker cells in mutants is not clear to me. Pacemaker cells have increased, not decreased resting membrane potentials, decreased Vmax (due to lower sodium channel activity), and unstable resting membrane potentials. The authors could analyse the electrophysiological properties of the pacemaker cells in wt and mutants (fgf13a-gfp label is perfect for that purpose), and also investigate the transcriptomes for differential expression of particular ion handling proteins known to be involved in the action potential morphology, membrane potential and intercellular conduction.

8) Line 217 "…slower conduction…" and line 219 "… more conduction system-like…" If anything, the conduction system shows very high conduction velocities. Perhaps refer to pacemaker properties. The remark regarding Vmax here is not clear. Vmax is low in pacemaker cells compared to atrial cells.

9) Line 242, transdifferentiation. See comment 4.

10) The authors assume the identified enhancer regulates nkx2-5, but do not provide evidence for this. The enhancer could be removed from the genome, for example. Did the authors analyse the topologically associated domain of nkx2-5 to see whether the enhancer is part of it? Is any interaction data available?

Please provide more detailed data regarding the ATAC-seq experiment. How many accessible sites, how many differential accessible sites, examples of known atrial accessible sites, UCSC browser views of a few examples to get an impression regarding data quality and signal-to-noise ratios. What is the correlation between RNAseq identified target genes and differential accessibility?

---

## [Author Response]

Essential revisions:As you will see, reviewers refer to a strong paper addressing patterning of the atrial chamber and a striking new mechanism of pacemaker cell restriction, but they also indicate that further work would be necessary for publication. In particular, three major issues have been raised.(1.A) The claim of transdifferentiation is currently not supported by the data.

This first comment is essentially the same as Essential Revision comment #2. Please see the response to Essential Revision comment #2 below for how we specifically addressed issues of transdifferentiation vs progenitor recruitment.

(1.B) The definition of AC/VC/PC requires clarification, as well as that of the boundary between chambers. The same definition should be used throughout the paper.

Thank you for this suggestion to help clarify our references to the specific cardiomyocyte populations. In the revised manuscript, we have now changed the references to cardiomyocyte populations examined, such as Amhc^+^/Vmhc^-^, so that they are defined primarily by their expression of the differentiation and transgenic markers used for the specific analysis. We then state our interpretations with respect to what that means regarding the atrial, atrioventricular canal, ventricular, and pacemaker cardiomyocyte populations. The boundaries between chambers and cardiomyocyte populations were determined by the expression of the indicated markers. The references to the cardiomyocyte populations with the designated markers have been changed throughout the text and figure legends, the main figures, and the figure supplements.

(1.C) The tracking of a limited number of markers (fgf13a:GFP transgene for example) to ascribe cell lineage or inconsistencies between markers (cells that take on ventricular gene expression in mutants have more of an atrial electrophysiological character) raise concerns about the conclusion of transdifferentiation.

a. We acknowledge that unfortunately there are a limited number of specific markers to use for some of the cardiomyocyte populations, in particular the pacemaker cardiomyocytes within the zebrafish hearts. This is something we have struggled with in our experiments. We did not use some established markers in that analysis, such as Bmp4, because they are expressed in the atrioventricular canal and pacemaker cells at the venous pole. This created some ambiguity to effects seen at later stages in the mutants due to the expression in both regions of the heart and the smaller atria. With other markers that should be expressed more specifically in the pacemaker cardiomyocytes, such as *shox* and *shox2*, there were technical issues in that the probes for both standard and HCR in situ hybridization, while marking the population, were not consistent or robust enough for us to perform quantitative analysis. Therefore, we primarily relied on immunohistochemistry for Isl1 and the *fgf13a:EGFP* reporter, because we found they were the most robust markers for the pacemaker cardiomyocytes currently available for analysis in the zebrafish hearts.

b. Regarding the electrophysiological characteristics (lines 248-280 of the revised manuscript), for the analysis of the action potentials we focused on regions that are closer to the middle of the morphological atrium where cardiomyocytes would express Amhc^+^ and *fgf13a*:EGFP^+^ and not express ventricular markers within the enlarged atrioventricular canal of *nr2f1a* mutant hearts based on our immunohistochemical analysis. Our data does show that this atrial region has electrophysiological properties more consistent with the central conduction system and not the WT atrial cardiomyocytes. The atrioventricular canal region also has altered conduction properties, as there is less slowing of conduction in the *nr2f1a* mutant atrioventricular canals. The lack of slowing of conduction at the atrioventricular canal of *nr2f1a* mutants is consistent with these cardiomyocytes taking on ventricular-like conduction properties and the gene expression changes we observe. Despite the improper conduction properties, as we showed previously larger endocardial cushions are still formed (Duong et al., 2018), supporting that the Amhc^+^/Vmhc^+^ cardiomyocytes still must have atrioventricular cardiomyocyte properties that promote valve formation. Please also see responses to Reviewer #3’s comments 6, 7, and 8.

(1.D) Inconsistencies between figures (1D versus 1J-J, 2F versus 4F, 1B and 2K, 1J-L versus S1 and 2G-I) should be resolved.

Thank you for point these out. We have tried to address each of the inconsistencies for the listed figures.

a. With respect to Figures 1D vs 1J-J, it appears the origin of this comment is Reviewer #2’s comment 1, “*Finally, the image shown in 1D does not appear to be congruent with the data in 1J-L – based on the image, I would expect to see more vmhc + CMs in the mutant atrium at 48 hpf*.” In considering this comment, it is not clear to us what Reviewer #2 found inconsistent about this data. Our data show there is an increase in total Vmhc^+^ cardiomyocytes in the *nr2f1a* mutant hearts compared to WT sibling hearts. However, the increase in total Vmhc^+^ cardiomyocytes comes from the combined Amhc^-^/Vmhc^+^ and Amhc^+^/Vmhc^+^ cardiomyocytes populations, which are marking ventricular and atrioventricular canal cardiomyocytes respectively. There is not an increase in the Vmhc+-only (Amhc^-^/Vmhc^+^) ventricular cardiomyocytes in the *nr2f1a* mutants relative to the WT control embryos. This is illustrated in Figure 1 —figure supplement 2 of the revised manuscript (Supplemental Figure 1 of the original manuscript), which shows there is not a statistically significant difference in the number of Vmhc^+^-only (Amhc^-^/Vmhc^+^) cardiomyocytes from these conditions from 48-96 hpf. To address this comment and better convey the expansion of Vmhc expression within the *nr2f1a* mutant hearts, we have provided the individual image channels in a revised Figure 1 —figure supplement 1 and also show the proportions of Vmhc^+^ cardiomyocyte populations in the hearts in Figure 1 —figure supplement 2D.

b. With respect to Figures 2F vs 4F, it appears this comment originated from Reviewer #2’s comment 3, “*The extent of fgf13a:GFP + cells in the atrium in nr2f1a mutants at 96hpf appears to differ markedly in Figures 2 (most of the cells are +) vs 4 (appears to be a lower proportion of the atrium as shown). How are chamber boundaries being defined in these analyses?*” We respectfully disagree that there is a significant difference in what is being shown between the effects in these images. In both Figure 2F and Figure 4F, the *fgf13a:EGFP* expression extends approximately half way across smaller atrium of the *nr2f1a* mutant hearts and overlaps significantly with Amhc^+^ cardiomyocytes. There can be variation in both the morphology of the hearts and the expansion of the *fgf13a:EGFP* expression in the *nr2f1a* mutants, with the representative heart in Figure 2F being more distended than Figure 4F. However, we are not basing the analysis and interpretations on cardiac or chamber morphology. We are basing the interpretation on the overlap of Amhc and *fgf13a*:EGFP expression. Importantly, our quantification of cardiomyocytes shows in Figure 2H that despite their morphology the *nr2f1a* mutant hearts have consistently more Amhc^+^/*fgf13a*:EGFP^+^ cardiomyocytes than sibling control embryos by 96 hpf. To address this comment and clarify our presentation of this data, we have revised Figure 4 to include the individual panels so that they are also showing Amhc expression within the hearts, which allows for better visualization of Amhc and *fgf13a*:EGFP expression within the atrium compared to the merged image in Figure 4F.

c. With respect to Figures 1B vs 2K, the RNA-seq in Figure 1B was performed on sorted *amhc*:EGFP^+^ cardiomyocytes at 48hpf. While Figure 2K was RT-qPCR for the respective genes from *myl7:EGFP^+^* (pan-cardiac) cardiomyocytes flow sorted at 96hpf. We performed the experiments this way as they are complementary analyses and provide insights into changes observed at the earlier stages of analysis that generated the hypotheses and those found to change at later stages of heart development following our temporal analysis. The sorting using a pan-cardiac marker was chosen because the smaller *amhc:EGFP*^+^ population observed at 96 hpf was technically difficult to sort and so we could better analyze changes in ventricular markers within the whole hearts. Figure 2K shows that by 96 hpf, the hearts overall have increased expression of selected ventricular and pacemakers markers, consistent with what we observe with the IHC shown in Figures 1 and 2.

d. With respect to Figures 1J-L vs S1, we do not understand what exactly could be inconsistent with these graphs. As we indicated in the Figure S1 legend of the original manuscript, the data shown in Figure S1 (Figure 1 —figure supplement 2 of the revised manuscript) is the same data shown in Figures 1J-L. These are not separate experiments. We separated out the individual cardiomyocytes populations in Figure 1 —figure supplement 2 to make it easier for the reader to observe the effects on the individual cardiomyocyte populations, as the stacked graphs may be difficult to easily observe some of the differences.

e. With respect to Figure 1J-L vs 2G-I, thank you for pointing this out. We understand why there many have been some confusion. There can be some slight experiment-to-experiment variability in the number of cardiomyocytes quantified within the hearts, which can be affected by a variety of factors, including the embryos, the IHC, and the microscope used. Importantly, despite slight differences in cardiomyocyte numbers for these different experiments, the trends on the cardiomyocyte populations quantified in the graphs between the original Figure 1J-L and Figure 2G-I were the same and did not affect the interpretation that there were decreased Amhc^+^ cardiomyocytes and increased Vmhc^+^ (ventricular) and *fgf13a*:EGFP^+^ (pacemaker) cardiomyocytes in the *nr2f1a* mutant hearts. However, to alleviate any confusion, we have replaced the graphs in Figure 2G-I with an experimental replicate showing the same trends, but the cardiomyocytes counts were more similar to those of Figure 1J-L. Again, the trends are the same from all the biological replicates and do not affect the interpretation regarding the changes in cardiomyocyte populations.

(1.E) An in-depth analysis of the ATAC-seq and RNA-seq data in wild-type and mutant atrial cells and comparison with epigenetic state data from other models would provide better insights into lineage changes.

We appreciate this comment and suggestion. Certainly, gaining a greater understanding of the level of conservation for the roles of Nr2fs within cardiomyocytes shown in this study is a goal that we have. However, it is not clear to us from the comment how these proposed comparisons should be done. For these types of comparisons, ATAC-seq and RNA-seq would need to be available from pretty similar cell types and collected in a similar manner, as chromatin state and expression can be very dynamic depending on factors such as developmental stage and cell type. Therefore, it is not clear to us that there is an available dataset we could use for this comparison to look at conserved changes in chromatin or gene expression, and even if there was, that it would immediately provide additional insight into the lineage changes. Another problem is that many cardiac enhancers are not highly conserved in the heart compared to other tissues (Blow et al., 2010). While the putative *nkx2.5* enhancer may not be highly conserved at the sequence level, we cannot rule out that is functionally conserved. However, this additional analysis would take significantly more time to perform. We have modified the text to specifically mention the lack of conservation of the putative *nkx2.5* enhancer element within the heart and that this may not be unusual, in the revised manuscript (lines 335-337).

(1.F) In addition, cell tracking is required to conclude transdifferentiation.

This comment addresses the same issue as Essential Revision comment 2. We have included additional experimental data in the revised manuscript to support that cardiomyocytes are transdifferentiating in the *nr2f1a* mutant hearts. Please see our response to Essential Revisions comment 2 below.

2) Alternative mechanisms through which pacemaker cells are expanded should be addressed experimentally: cell death, cell proliferation, or progenitor recruitment.

A major concern from the reviewers was that they felt we had not ruled out the possibility that the additional pacemaker cardiomyocytes in the *nr2f1a* mutant hearts could be added from another source, such as through progenitor recruitment and newly-differentiated pacemaker cardiomyocytes at the venous pole, and were not from the transdifferentiation of atrial (Amhc^+^) cardiomyocytes. We did mention in the original manuscript that we did not see effects on cardiomyocyte proliferation or death within the hearts of *nr2f1a* mutants. However, we did not show this data as it was negative. Furthermore, our data in Figure 1L show that the *nr2f1a* mutant hearts effectively do not grow from 48 to 96 hpf. Therefore, it is unlikely the expansion of pacemaker cardiomyocyte gene expression within the *nr2f1a* mutant hearts could be explained by being from newly-added cardiomyocytes. To explain the pacemaker expansion being from newly-differentiating pacemaker cardiomyocytes with a static number of total cardiomyocytes, one would have to posit there is a treadmilling effect with atrial cardiomyocytes dying at a significant rate and being replaced by newly-differentiating pacemaker cardiomyocytes. To specifically address the issue of progenitor recruitment/newly-differentiating cardiomyocytes vs transdifferentiation, in the revised manuscript we have included the data examining cell death with active Caspase 3 and cell proliferation using both pHH3 and EdU. We also performed analysis of the temporal addition of newly-differentiating cardiomyocytes within the *nr2f1a* mutant hearts with the previous reported Tg(*myl7:NLS-KikGR)* transgene. Altogether, these data support that the Amhc^+^ cardiomyocytes in the venous pole of the atrium are acquiring pacemaker gene expression. The text of the revised manuscript now refers to these experiments and the new supplemental figures (lines 189-198 and 229-247, and Figure 1 —figure supplement 3, Figure 1 —figure supplement 4, and Figure 2 —figure supplement 3).

(3.A) The identification of nkx2.5 as a target of nr2f1a to repress pacemaker cells needs strengthening. Please explain how nrf2f1a is envisioned to act as a suppressor of pacemaker phenotype (atrial-specific activator of nkx2-5) as the gene is expressed in pacemaker cells and in the posterior second heart field progenitors the atria and sinus venosus (including pacemaker cells) derive from.

We agree that we do not yet understand how Nr2f1a mechanistically promotes/maintains Nkx2.5 expression in venous atrial cardiomyocytes and represses pacemaker identity in atrial cardiomyocytes. Since Nr2f1a is expressed throughout atrial cardiomyocytes (Figure 6 —figure supplement 3), including pacemaker cardiomyocytes, we hypothesize that Nr2f1a must interact with other factors and signaling to provide these regional effects. The identification of these factors and signaling extends beyond this manuscript. We specifically discuss and state this hypothesis in the Discussion of the revised manuscript (lines 436-443).

(3.B) Maintenance of nkx2.5 expression by nr2f1a at the venous pole is incongruent with the atrial to ventricular shifts observed. More details on nr2f1a expression and co-expression with nkx2.5 and pacemaker markers would be useful.

a. Unfortunately, it is not clear to us how maintenance of *nkx2.5* expression is incongruent with the atrial to ventricular shifts. In *nkx2.5* zebrafish mutants, the hearts fail to maintain ventricular identity and transition to atrial and pacemaker cardiomyocyte identity (Colombo et al., 2018). As Nkx2.5 expression is not lost in the atrioventricular canal of the *nr2f1a* mutants, but the more venous atrial cardiomyocytes, we would not predict that the ability to promote or maintain ventricular identity would be affected in the *nr2f1a* mutant hearts.

b. To address the comment about co-expression of Nr2f1a and Nkx2.5, we have added immunohistochemistry of Nr2f1a and Nkx2.5 with pacemaker markers. However, they cannot be examined together as the antibodies were both generated in rabbits. The data of the co-expression are mentioned in the Results and Discussion and supplemental figures of the revised manuscript (lines 287-291 and 436-443 Figure 4 —figure supplement 4 and Figure 6 —figure supplement 3).

(3.C) The authors assume the identified enhancer regulates nkx2-5, but do not provide evidence for this.

In the revised manuscript, we have examined the function of the putative *nkx2.5* enhancer using CRISPR/Cas9 to delete it. We find that deletion of the putative *nkx2.5* enhancer in embryos leads to a loss of Nkx2.5^+^ cardiomyocytes and an expansion of the *fgf13a:*EGFP reporter in atrial cardiomyocytes. This new experiment has been described in the Results (lines 348-364) and included in a revised Figure 6 and new Figure 6 —figure supplement 2 of the revised manuscript.

(3.D) Overexpression of nkx2.5 not completely suppresses but delays the appearance of pacemaker cells. Examination of nkx2.5 rescue should be extended at a functional level (cardiac function and electrophysiology).

We assessed heart rate following induction of Nkx2.5 at the 20 somite stage in *nr2f1a* mutants at 96 hpf. Despite the strong effects on the *fgf13*:EGFP reporter in *nr2f1a* mutants, this does not restore heart rate at that stage. One reason for the failure to rescue heart rate could be that at this stage the pacemaker cardiomyocytes have already started to expand. Furthermore, this result is consistent with the hypothesis that additional factors besides Nkx2.5 are needed to fully rescue atrial cardiomyocyte and restrict pacemaker cardiomyocyte identities within the venous pole of the atrium. This data is now mentioned in the Results (lines 314-319) and included in a new Figure 5 —figure supplement 1 of the revised manuscript.

(3.E) Detailed comments of the reviewers are below. Please take into account all comments and provide more explanations on the methods as detailed by the reviewers. If you are not able to address experimentally some reviewer comments within the revision period, please discuss the limitations or alternative hypotheses in the manuscript.

We have responded to all the comments and comments not addressed in the Essential Revisions below. We have expanded the Methods as requested by the reviewers, and we have included the Methods used for new experiments added to the revised manuscript.

Reviewer #1 (Recommendations for the authors):– Please provide a clear indication of the biological system under investigation in the title.

We have revised the title to indicate that the studies use zebrafish.

– Abbreviations complicate reading.

We have tried to limit the use of abbreviations for the cardiomyocyte populations in the revised manuscript. This was in large part achieved through changing how the populations of cardiomyocytes are referenced. When we did use abbreviations, we made sure they are clearly defined.

– How fluorescence of Nkx2.5-EGFP is distinguished over the Et(fgf13a:EGFP) transgene?

As we indicated in the Methods of the original manuscript, prior to the heat-shock, we first sorted embryos for those that were transgenic for the *fgf13a*:EGFP expression based on its expression in the epidermis. Following the heat-shock, transgenic and non-transgenic *hsp70l:nkx2.5-EGFP* embryos were sorted based on induced Nkx2.5-EGFP expression, which shortly after the heat-shock is significantly brighter than the transgenic *fgf13a*:EGFP expression. In the revised manuscript, we have tried to clarify how we distinguished the expression of these transgenes with a statement in the Results (lines 301-303) and in the Methods (lines 584-588). In the initial manuscript, we unintentionally omitted that the genotypes of all embryos were subsequently confirmed with PCR genotyping following imaging for the presence or absence of the *hsp70l:nkx2.5-EGFP* transgene. This is now also included in the revised Methods (lines 591-593).

– Was cell counting done in 3D or in 2D in a projection image? Can a partial cell count in 2D explain variations between Fig2G+H and 1J? Please clarify the method section.

The cardiomyocytes were counted using MaxIP projections from confocal images of the whole hearts. We do not think partial counts could explain variation between these figures. It is more likely some experiment-to-experiment variation that can occur. Please see our response to Essential Revisions comment 1D for additional details of how we addressed this issue in the revised manuscript.

– Please complete the criteria of differential expression in the RNA-seq (p-value or adjusted p-value ? cut-off ?).

This data should have been included in Supplementary file 1. There is now a sheet with all the RNA-seq data included in a revised Supplementary file 1.

– "ectopic expression" (line 139) is not correct since amhc positive cells normally do express ventricular and pacemaker markers.

Thank you. We have modified this sentence (line 150 of the revised manuscript).

– Two mutant lines are described in the method section. Which line is used for each experiment? How have they been validated for the loss of nr2f1a expression?

The *nr2f1a^el512^* and *nr2f1a^ci1009^* mutant alleles lines were reported previously (Duong et al., 2018). The *nr2f1a^ci1017^ allele* was verified phenotypically, through complementation tests with the previous lines, and Western blots for Nr2f1a. As indicated in the text, additional details of the allele are in another manuscript that will be published shortly. As the *nr2f1a* mutant cardiac phenotype was published previously, it was not necessary to belabor the point of the additional alleles. The *nr2f1a^ci1017^* allele was used for the RNA and ATAC-seq experiments. The *nr2f1a^el512^* and the *nr2f1a^ci1009^* alleles were used for the temporal differentiation assay with the *myl7:KikGR*. All the rest of the experiments in the manuscript were performed with the *nr2f1a^ci1009^* allele*.* There was no experimental rationale for using these lines for different experiments, other than they had the necessary transgenes to perform desired experiments crossed into them.

– How are vmhc and AVC markers in the transcriptomic dataset?

*Vmhc* (*myh7*) for some reason was not found in the RNA-seq data sets from sorted *amhc*:EGFP^+^ cardiomyocytes. However, other markers of ventricular identity were, as we indicated. We know from previous analysis (Duong et al., 2018) and RT-qPCR shown in Figure 2K that *vmhc* expression is increased in the whole hearts. Despite the trends we observe, it could reflect that a limited number of these cardiomyocytes were captured due to lower *amhc* expression in these cardiomyocytes from *nr2f1a* mutant hearts and the expression didn’t meet the defined cut-offs. Furthermore, it is difficult to distinguish some of the atrioventricular canal markers from pacemaker markers at this stage. Again, as we reported previously, when examining the heart atrioventricular canal markers are also expanded (Duong et al., 2018).

– Provide single-channel images for Figure 1C-H, 2A-F to clarify marker overlap.

Single-channel images have been provided for Figures 1 and 2 in Figure 1 —figure supplement 1 and Figure 2 —figure supplement 1 of the revised manuscript.

Reviewer #2 (Recommendations for the authors):1. Figure 1: the overlap in Amhc and Vmhc should be more clearly shown. As there appears to be a significant overlap by 48 hpf (1D), what do these markers show at earlier time points? Finally, the image shown in 1D does not appear to be congruent with the data in 1J-L – based on the image, I would expect to see more vmhc + CMs in the mutant atrium at 48 hpf.

a. We have provided images of hearts at an earlier time points (30 hpf). These show that there is actually significant overlap of Vmhc and Amhc at these earlier stages in the heart tube, which to our knowledge has not really been reported previously at the shown stage. Importantly, we did not find a difference in the overlap of Vmhc and Amhc between WT and *nr2f1a* mutant hearts at this stage. This suggests that Nr2f1a is required to refine the number of Vmhc and Amhc cardiomyocytes in the atrioventricular canal, consistent with our previous report (Duong et al. 2018). This new data and related interpretations are now described in the Results and Discussion of the revised manuscript (lines 160-162 and 385-390) and in new panels included in Figure 1- supplement 1.

b. We have tried to address the issue of inconsistency between Figure 1D and 1J-L in the response to Essential Revisions comment 1D.

2. Line 168: The cell death and proliferation data are important to argue it is a fate conversion vs loss/expansion of populations that is occurring. It should be shown, or some form of fate tracing (photoconversion, etc.) should be used if statements like those in lines 174/5 are to be made.

Thanks you. We have provided the cardiomyocyte death and proliferation analysis, and additional lineage tracing/temporal differentiation assay to examine newly-differentiating cells in the *nr2f1a* mutant hearts to address the issue of fate conversion/transdifferentiation. Please see our response to Essential Revision comment 2 for additional details of how we addressed this issue.

3. Figure 4: The extent of fgf13a:GFP + cells in the atrium in nr2f1a mutants at 96hpf appears to differ markedly in Figures 2 (most of the cells are +) vs 4 (appears to be a lower proportion of the atrium as shown). How are chamber boundaries being defined in these analyses?

Please see our response to Essential Revision comment 1D for how we addressed this issue.

4. Figure 5: Somewhat confusing – how are new PCs made from 48 to 96hpf if Nkx2.5 is still present? ACs are still lost in "rescued" hearts – presumably, these now make VCs? Importantly, does the nkx2.5 transgene rescue/normalize heart rate or conduction? It is important to distinguish an effect on the fgf13a transgene (which is convincing) versus a more important effect on PC and AC fate and FUNCTION is reflected in these results.

a. It is difficult to address this issue because it is not clear to us from this comment what exactly Reviewer #2 finds confusing. Normally, Nkx2.5 is expressed in virtually all cardiomyocytes within the zebrafish heart except for pacemaker cardiomyocytes at the venous pole of the heart. This is now shown in Figure 4 —figure supplement 1 of the revised manuscript. In the *nr2f1a* mutants, Nkx2.5 is not completely lost throughout the heart. It just recedes from the venous pole of the heart, which corresponds with the new marker expression for the pacemaker cardiomyocytes (Figure 4). As Nkx2.5 is still expressed throughout the heart in the mutants, we would not expect to observe difference in the effect on ventricular cardiomyocytes.

b. In the “rescued” hearts where Nkx2.5-EGFP is induced, it is not clear that Nkx2.5-EGFP is still visible at the later stages following the heat-shock. The previous characterization of the *hsp70l:nkx2.5-EGFP* line showed that Nkx2.5-EGFP was no longer detectable by 10 hrs after the heat shock (George et al., 2015). While inducing Nkx2.5-EGFP at the 20 somite stage is sufficient to rescue *nkx2.5* mutant embryos to adulthood (George et al., 2015), it does not mean it is perpetually expressed following the heat shock at the stages examined in the analysis. Moreover, we do not see the Nkx2.5-EGFP at 48 through 96 hpf following the heat-shock at 20 somites. Similar to the *nkx2.5* mutant rescue, we see maximal repression of the pacemaker reporter (*fgf13a:EGFP*) when induced at the 20 somite stage. These results suggest that even though we observe the regression of Nkx2.5 at later stages in the mutants, Nkx2.5 and Nr2f1a complexes must be required earlier to establish these boundaries. We include this in the Discussion of the revised manuscript (lines 421-425 and 429).

c. We did assess rescue of heart rate following Nkx2.5-EGFP induction as requested. Please see our response to Essential Revisions comment 3.D.

Reviewer #3 (Recommendations for the authors):1) Please provide more details in figure 1 regarding the RNAseq analysis. Why did the authors use a cut-off of >2 fold difference rather than significance? Could they provide an MA plot or so to show distributions of FC and p values of all transcripts detected? How many transcripts were differentially expressed? What kind of biological processes were changed?

We did not present the RNA-seq data this way as we did not think it provided additional information that was necessary. It served as the basis for additional in vivo investigation of a hypothesis in the embryos. All of the RNA-seq data are now included in a sheet in Supplement File 1. They were unintentionally omitted from the excel sheet originally submitted. The expression can be represented in different ways. We felt the differences were best illustrated using fold difference changes that were found in this data set.

2) In figure 1B the behavior of pacemaker markers is inconsistent. Please provide data or references showing the expression of these markers in fish pacemaker cells (e.g. I did not find these data for tbx3a etc.). Furthermore, central pacemaker markers like isl1a (Isl1 homologue) and shox2a (Shox2 homologue) are downregulated in mutants? That is unexpected, and also inconsistent with the data given in figure 2K.

a. Please see our response to Essential Revision comment 1D.

b. References to the expression of the pacemaker marker genes in the zebrafish have been added to the revised manuscript (lines 147,148, 207-209).

3) Arterial pole of the atrium is confusing, as arterial pole usually refers to the actual arterial pole of the heart (distal/downstream portion of the ventricle).

We welcome suggestions for what would be the best way to refer to this anatomical position in the context of our study as we would like to be as clear as possible. Given the changes in heart morphology and markers within the different regions of the *nr2f1a* mutant hearts, it was not clear how to best refer to this portion of the atrium, which is not often specifically referred to as most studies have not had reason to refer to these different regions within the zebrafish atrium. Therefore, it seemed clearest to us to use arterial and venous as relative anatomical position markers with respect to the direction of blood flow in the heart, as is done with respect to the whole hearts in fish and mammals, and contrast these positions with our references to the pacemaker cardiomyocytes at the venous pole of the heart/atrium. In the Abstract and Introduction of the revised manuscript, we have also defined arterial and venous as the outflow and inflow portion of the atrium to help clarify this point (lines 27, 29, 110, and 114).

4) Please explain how the authors envision nrf2f1a/Nrf2f2 acts as a suppressor of pacemaker phenotype (atrial-specific activator of nkx2-5) as the gene is expressed in pacemaker cells and in the posterior second heart field progenitors the atria and sinus venosus (including pacemaker cells) derive from.

Please see our responses to Essential Revisions comments 3.A and 3.B, Reviewer #1’s major comment 3.

5) The cell numbers seem inconsistent between the different figures. In suppl Figure 1, there are about 75 amhc+ cells in wt, and 20 or so in mutants. In figure 1, there are close to 100 amhc+ cells in wt, and about 50 in mutants? In figure 2 there are 60 amhc+ cells in wt, and 40, 20 and 10 amhc+ cells in mutants at subsequent stages. If one adds the amhc+ and fgf13a-gfp+ cells, the numbers are 40, 30, 20, respectively (figure 2c). Which numbers are correct? Is the number of amhc+ cells very low in mutants from stage 48 hpf onwards, without changing, or is there a decrease during development?A more general issue related to the cell number issues, the data can be explained by transdifferentiation of atrial cells into pacemaker cells, as the authors seem to favor, or by reduced addition of cardiomyocytes fated to become atrial, while the normal atrial-pacemaker demarcation programs are still active, or by reduced atrial differentiation. The latter 2 options are consistent with the data of suppl Figure 1. The authors should resolve this, and provide additional support for the trans-differentiation hypothesis. Thus, does a fraction of cells expressing pacemaker markers derive from atrial marker expressing cells in mutants, or is there an increased population of "de novo" pacemaker cells (not previously atrial marker positive) in the venous pole and a decreased population of atrial cells because of defective recruitment from progenitors or defective atrial differentiation.

a. Please see our response to Essential Revisions comment 1.D for how we addressed concerns of the inconsistencies.

b. Please see our response to Essential Revisions comment 2 for how we experimentally addressed concerns of transdifferentiation vs. new addition of cardiomyocytes.

6) Line 201: "Thus, we would predict…" Why would the heart rate be predicted to decrease? More pacemaker cells in mutants may also result in increased automaticity and increased spontaneous depolarizations.

Thank you for the comment. The reviewer is correct. One might readily imagine ways in which increased numbers of pacemaker cells could lead to any number of heart rate outcomes depending on their cell autonomous features, their capacitance, and other biophysical properties as well as particular patterns of intercellular coupling. In our description of the anticipated heart rate findings, we neglected to fully outline our rationale. Given the typical intrinsic automaticity of a coupled cellular mass is a function of the total electrical capacitance of that mass, we simply hypothesized that the intrinsic heart rate might be slower as the mass of coupled pacemaker cells increased. Clearly, this assumes that cellular characteristics and coupling are unchanged, but as these properties were also measured directly in the work described we did not elaborate on these elements at this stage. We have revised the text to be more explicit on this point (lines 252-255).

7) The relation between hyperpolarization, increased repolarization times and increased numbers of pacemaker cells in mutants is not clear to me. Pacemaker cells have increased, not decreased resting membrane potentials, decreased Vmax (due to lower sodium channel activity), and unstable resting membrane potentials. The authors could analyse the electrophysiological properties of the pacemaker cells in wt and mutants (fgf13a-gfp label is perfect for that purpose), and also investigate the transcriptomes for differential expression of particular ion handling proteins known to be involved in the action potential morphology, membrane potential and intercellular conduction.

We apologize if the relationship between the various cellular parameters, myocardial physiology, and cell identity markers was not clear. Our intent was to convey that the atria in the *nr2f1a* mutants exhibit intermediate electrophysiologic characteristics consistent with cardiomyocytes of the central conduction system, although we cannot readily classify them as pacemaker, atrial, or ventricular cardiomyocytes. While we cannot perform the analysis in the *fgf13a:EGFP* transgenic fish, we did analyze the electrophysiology in different regions of the atria that would be consistent with them having *fgf13a*:EGFP^+^ and *fgf13a:*EGFP*^-^* cardiomyocytes in our studies and did not identify discrete physiological boundaries. This intermediate state which we identified in the *nr2f1a^-/-^* atria exhibits atrial Vmax and ventricular repolarization with hyperpolarization of the resting membrane potential and no significant changes in phase 4 depolarization. These features and the slowed conduction observed suggest cellular electrophysiology and cell-cell interactions most consistent with the central conduction system. This is of importance for several reasons not least the known gradients of function between atrial cardiomyocyte and central conduction system observed from mid-atrium distally in humans. We have clarified the relevant text in the Introduction, Results, and Discussion (lines 118, 252-255, 266, 267, 272, 273, 408).

8) Line 217 "…slower conduction…" and line 219 "… more conduction system-like…" If anything, the conduction system shows very high conduction velocities. Perhaps refer to pacemaker properties. The remark regarding Vmax here is not clear. Vmax is low in pacemaker cells compared to atrial cells.

We apologize that we did not make it clear that we were referring not to the ventricular (His-Purkinje) conduction system but rather to the central conduction system whose biology we believe these experiments are helping to elucidate. We have clarified the relevant text of the revised manuscript (lines 118, 252-255, 266, 267, 272, 273, 408).

9) Line 242, transdifferentiation. See comment 4.

Please see our response to Essential Revisions comment 2 for how we experimentally addressed concerns of transdifferentiation vs. new addition of cardiomyocytes.

10) The authors assume the identified enhancer regulates nkx2-5, but do not provide evidence for this. The enhancer could be removed from the genome, for example. Did the authors analyse the topologically associated domain of nkx2-5 to see whether the enhancer is part of it? Is any interaction data available?Please provide more detailed data regarding the ATAC-seq experiment. How many accessible sites, how many differential accessible sites, examples of known atrial accessible sites, UCSC browser views of a few examples to get an impression regarding data quality and signal-to-noise ratios. What is the correlation between RNAseq identified target genes and differential accessibility?

a. Thank you for the suggestion. We have provided experimental analysis of the putative *nkx2.5* enhancer. Please see response to Essential Revisions comment 3.C.

b. While it would be ideal and we would have loved to analyze the topologically association of chromatin in these zebrafish cardiomyocytes, presently it is not possible to perform the techniques, such as HI-C, to analyze chromatin conformation in these cells. Therefore, we cannot presently directly address this comment experimentally.

c. The ATAC-seq data and their relationship to changes in gene expression from the RNA-seq are summarized in a sheet in Supplementary file 1. Examples of UCSC browser views for select genes have now been added to a new Figure 6 —figure supplement 1 of the revised manuscript as requested to show the quality of the data. Additionally, Bigwig files were submitted to GEO.